# RouterDC: Query-Based Router by Dual Contrastive Learning for Assembling Large Language Models

**Shuhao Chen**[1, ⋆], **Weisen Jiang**[1, 2, ⋆], **Baijiong Lin**[3], **James T. Kwok**[2], **Yu Zhang**[1, †]
[1]Southern University of Science and Technology
[2]The Hong Kong University of Science and Technology
[3]The Hong Kong University of Science and Technology (Guangzhou)
`12232388@mail.sustech.edu.cn, jamesk@cse.ust.hk`
`{waysonkong, bj.lin.email, yu.zhang.ust}@gmail.com`

## Abstract

Recent works show that assembling multiple off-the-shelf large language models (LLMs) can harness their complementary abilities. To achieve this, routing is a promising method, which learns a router to select the most suitable LLM for each query. However, existing routing models are ineffective when multiple LLMs perform well for a query. To address this problem, in this paper, we propose a method called query-based Router by Dual Contrastive learning (RouterDC). The RouterDC model, which consists of an encoder and LLM embeddings, is trained by two proposed contrastive losses (sample-LLM and sample-sample losses). Experimental results show that RouterDC is effective in assembling LLMs and largely outperforms individual top-performing LLMs as well as existing routing methods on both in-distribution (+2.76%) and out-of-distribution (+1.90%) tasks. The source code is available at `https://github.com/shuhao02/RouterDC`.

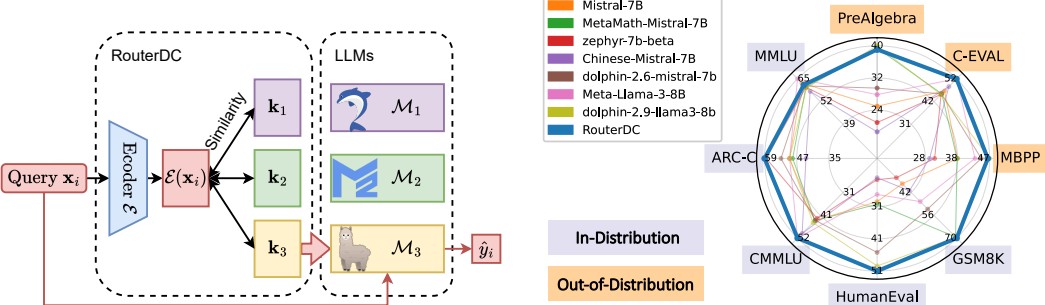

Figure 1: The inference pipeline of RouterDC. The encoder $\mathcal{E}$ and the LLM embeddings $\mathbf{k}$'s are trainable parameters, while the LLMs are frozen.

Figure 2: Testing accuracy of candidate LLMs and our RouterDC on in-distribution and out-of-distribution tasks.

## 1 Introduction

Large language models (LLMs) have demonstrated proficient capabilities across various tasks. Many LLMs are publicly available online, such as Mistral [24], LLaMA-2 [46], and LLaMA-3 [45]. Those LLMs have been further fine-tuned to be generalists or specialists. For example, MetaMath [54] excels in solving mathematical reasoning problems. Since those LLMs are pre-trained or fine-tuned with various data, they typically exhibit varying strengths and weaknesses across different tasks

---

⋆Equal contribution    †Corresponding author

38th Conference on Neural Information Processing Systems (NeurIPS 2024).

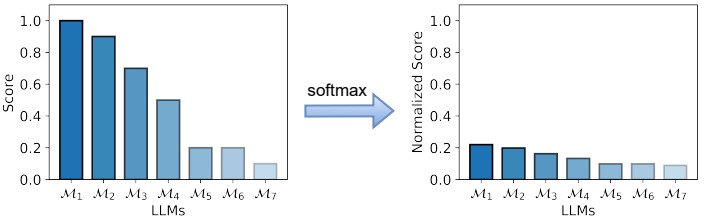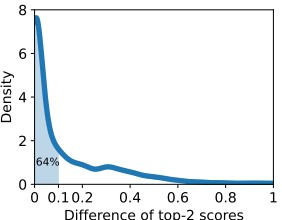

Figure 3: Score distributions of LLMs on an example query (w/ or w/o normalization).

Figure 4: Distribution of the score difference between the top two LLMs.

[25, 33]. Therefore, assembling multiple off-the-shelf LLMs can harness their complementary abilities, resulting in better performance than relying on a single LLM.

LLM ensembling is a straightforward method to assemble LLMs, which feeds the query to all candidate LLMs and merges all outputs into a final answer by majority voting [31, 51] or pairwise ranking [25]. However, ensembling is computationally prohibitive as it requires generating outputs with all candidate LLMs during inference. To tackle this issue, recent works [43, 33, 44] propose to learn a router to select a suitable LLM for each query. During inference, routing is much more efficient than ensembling as it only needs to perform inference on the selected LLM.

The current state-of-the-art routing method is ZOOTER [33]. To train a router, ZOOTER scores the outputs of candidate LLMs as the supervision signal via an off-the-shelf reward model and then learns the router by minimizing the Kullback-Leibler divergence [29] between the selection probability from the router and the softmax normalized score. However, this loss is inappropriate when multiple LLMs perform well for a query. Figure 3 shows the scores of seven LLMs for an example query, where the top three LLMs have significantly higher scores than the bottom three LLMs. After the softmax normalization, the scores are small, leading the router to generate small probabilities on the top LLMs. Moreover, the normalized score tends to be uniform, which is not a strong supervision signal for learning the router. Figure 4 shows that the score difference between the top two LLMs is usually tiny (under the experimental setting in Section 4.1), indicating that the loss used in ZOOTER is inappropriate.

In this paper, we propose a query-based **Router** by **D**ual **C**ontrastive learning (RouterDC). The RouterDC consists of an encoder, whose architecture is a small language model, and learnable LLM embeddings for candidate LLMs. For each query, we first score the candidate LLMs by comparing their predictions with the gold label. Instead of directly aligning the score distribution, we leverage the score to choose the top-performing and bottom-performing LLMs and then propose a *sample-LLM contrastive loss* to pull the query embedding (extracted by the encoder) close to the embeddings of top LLMs while pushing far away from the embeddings of bottom LLMs. Based on this loss, our RouterDC could equally select one of top-performing LLMs for a query and hence alleviate the shortcoming of ZOOTER introduced previously. We empirically observe that training the router using the sample-LLM contrastive loss alone is not stable as similar queries can have dissimilar embeddings and be assigned to different LLMs. To improve the training stability, we cluster all the training queries into multiple groups and design a *sample-sample contrastive loss* to maximize the similarity between queries in the same group while minimizing the similarity between queries from different groups.

We conduct experiments on challenging reasoning tasks (language understanding, code generation, and mathematical reasoning) to evaluate the proposed RouterDC in both in-distribution and out-of-distribution settings. Empirical results show that RouterDC can harness the complementary potentials of LLMs, achieving state-of-the-art performance. Moreover, RouterDC outperforms existing routing methods by a large margin, showing that the proposed two contrastive losses are more beneficial for training the RouterDC.

Our contributions are summarized as follows. (i) We propose a *novel framework* to learn a router to select the suitable LLM for each query by dual contrastive learning, which consists of sample-LLM and sample-sample contrastive losses; (ii) The proposed RouterDC is *parameter-efficient* (with fewer than 100M parameters) and *computation-efficient* (without backpropagating the gradients through LLMs) in training. Moreover, RouterDC is also efficient in inference ($6\times$ faster than Voting) as it only requires computation cost for the selected LLM and negligible cost for the router;

(iii) Experimental results show that RouterDC *effectively assembles LLMs* and outperforms individual top-performing LLMs as well as existing routing methods on both in-distribution (+2.76%) and out-of-distribution (+1.90%) tasks.

## 2   Related Work

**Large Language Models (LLMs).** LLMs have achieved great success in natural language processing and many foundation models have been released online [24, 46, 45, 27]. Many prior works [54, 47, 58, 9, 8, 41, 20, 34, 52] focus on fine-tuning those foundation models to obtain specialized LLMs for solving versatile tasks, for example, language understanding [58, 20, 26], code generation [41, 8], and mathematical reasoning [54, 34]. In this paper, we study the problem of assembling LLMs to harness their strengths by a router.

**LLM Ensembling.** The goal of LLM ensembling is to leverage multiple LLMs to boost performance compared with a single model across various downstream tasks. Voting [31, 51] is a simple but effective ensemble method. Jiang et al. [25] further propose PairRanker and GenFuser to generate an improved output from the outputs of all LLMs, which needs to call LLMs $\mathcal{O}(T^2)$ times with $T$ as the number of LLMs. LLM cascading [2, 13, 38, 55] query a list of LLMs (whose capacity depends on the model size) sequentially until an LLM's output is satisfied (i.e., having a significantly high confidence score), which is returned as the final output. Fusion of Experts [50] concatenates all LLMs outputs to build the final output and casts it as a supervised learning problem. Unlike the aforementioned ensembling methods which require querying the LLMs at least $\mathcal{O}(T)$ times in inference, our RouterDC is much more efficient as it only needs to call the selected LLM once.

**LLM Routing.** LLM routing aims to select the most suitable model for a query without calling all LLMs. Many works have been proposed to design an effective routing strategy. Shnitzer et al. [43] propose a collection of binary classifiers to evaluate the correctness of each LLM. Lu et al. [33] propose ZOOTER to align a router with the supervision from the reward model. LoraRetriever [56] propose a task-wise router to select the LLM by predicting the task identity of the query. Srivatsa et al. [44] explore the routing ability using both classifier-based and clustering-based approaches. The aforementioned methods neglect the fact that multiple LLMs may be well-suited to answer a single query. Ding et al. [10] design a cost-effective router for two LLMs (a small LLM and a large one). In contrast, the proposed RouterDC can be used for multiple LLMs simultaneously.

**Contrastive Learning.** Contrastive learning learns effective representations by distinguishing between similar and dissimilar pairs of data points. It has been widely used in various tasks, such as visual representation learning [4, 14], sentence representation leaning [12, 53, 42], and vision-language alignment [39, 59]. In this paper, we propose two contrastive losses to learn the RouterDC for assembling LLMs.

## 3   Methodology

In this section, we propose RouterDC, a framework for learning a query-based router to assemble LLMs. An overview is illustrated in Figure 1. We introduce the problem of router learning in Section 3.1 and design a scoring method to measure the performance of LLMs on each training query (Section 3.2). Next, we propose two contrastive losses to train the router, including a sample-LLM contrastive loss for learning the routing strategy (Section 3.3) and a sample-sample contrastive loss for improving training stability (Section 3.4). The training and inference procedures are provided in Algorithm 1.

### 3.1   Problem Formulation

Consider a set of LLMs $\{\mathcal{M}_t : t = 1, \ldots, T\}$ and a training set $\mathcal{D}_{\text{train}} = \{(\mathbf{x}_i, y_i) : i = 1, \ldots, n\}$, where $\mathbf{x}_i$ is a query (i.e., question) and $y_i$ is its answer (i.e., ground truth). Usually, no single LLM is universally suitable for all queries in $\mathcal{D}_{\text{train}}$. Moreover, LLMs are diverse and have different architectures (e.g., Mistral-based [24], LLaMA-based [45]), making it infeasible to merge all LLMs into a single model [36, 22, 28]. In this paper, we study the problem of assembling LLMs by learning a router to select the suitable LLM for each query. The router takes $\mathbf{x}$ as input and produces the probability distribution of $T$ LLMs being selected. As training and testing queries may come from

different data distributions, the learned router is expected to generalize well on both in-distribution and out-of-distribution scenarios.

## 3.2 Scoring

To learn the router, we need to design a scoring method to assess the performance of LLMs on queries. For an *open-ended* generation query $\mathbf{x}_i$ (requiring a long answer, e.g., GSM8K [7], with an example shown in Example 1), one can directly compare the ground truth $y_i$ with the output of the LLM $\hat{y}_i^{(t)} = \mathcal{M}_t(\mathbf{x}_i)$ generated by greedy decoding. Though greedy decoding is simple and efficient, its inherent shortsightedness often prevents it from discovering the optimal solution. Conversely, sampling, like beam sampling [49], is an advanced approach that is widely used in practice as it explores multiple alternatives in the search space, potentially leading to better results. We repeatedly feed the query $\mathbf{x}_i$ to the LLM $\mathcal{M}_t$ $M$ times to obtain outputs $\{\hat{y}_{i,m}^{(t)} : m = 1, \ldots, M\}$. Then, we define the score of LLM $\mathcal{M}_t$ on the query $\mathbf{x}_i$ as:

$$s_i^{(t)} = \frac{1}{M} \sum_{m=1}^{M} \text{evaluate}(\hat{y}_{i,m}^{(t)}, y_i), \tag{1}$$

where $\text{evaluate}(\hat{y}, y)$ gives 1 if the prediction $\hat{y}$ is correct otherwise 0.

For a *multiple-choice question* $\mathbf{x}_i$ with an option set $\mathcal{A}_i$ (e.g., MMLU [17], as an example shown in Example 1), sampling is unnecessary as we can simply define the score based on the probability of options, i.e.,

$$s_i^{(t)} = \begin{cases} \frac{\mathbb{P}_{\mathcal{M}_t}(\hat{y}_i^{(t)}|\mathbf{x}_i)}{\sum_{a \in \mathcal{A}_i} \mathbb{P}_{\mathcal{M}_t}(a|\mathbf{x}_i)} & \text{if} \quad \hat{y}_i^{(t)} = y_i \\ 0 & \text{otherwise} \end{cases} \tag{2}$$

where $\mathbb{P}_{\mathcal{M}_t}(a|\mathbf{x}_i)$ is the probability of option $a$ predicted by the LLM $\mathcal{M}_t$. According to Eq. (2), when the LLM $\mathcal{M}_t$ outputs a correct option (i.e., $\hat{y}_i^{(t)} = y_i$), we normalize the probability to make it comparable across different LLMs, which will be used in Section 3.3; When the LLM $\mathcal{M}_t$ generates a wrong option, $s_i^{(t)}$ is set to 0 to punish $\mathcal{M}_t$ for $\mathbf{x}_i$. Based on the scores $\{s_i^{(t)} : t = 1, \ldots, T\}$, we introduce a sample-LLM contrastive loss in the next section.

---

**Example 1**

An *open-ended* question from GSM8K [7]:
**Question:** Tim has 30 less apples than Martha, and Harry has half as many apples as Tim. If Martha has 68 apples, how many apples does Harry have?
**Answer:** | Tim has 68-30 = 68-30=38 apples. Harry has 38/2 = 38/2=19 apples. #### 19 |

A *multiple-choice* question from MMLU [17]:
**Question:** An object is placed 100cm from a plane mirror. How far is the image from the object?
**Options:** A. 50cm    B. 100cm    C. 200cm    D. 300cm
**Answer:** | C |

---

## 3.3 Sample-LLM Contrastive Loss

As illustrated in Figure 1, the proposed RouterDC consists of an encoder $\mathcal{E}(\mathbf{x}; \mathbf{w})$ parameterized by $\mathbf{w}$ (where in our experiments $\mathcal{E}(\mathbf{x}; \mathbf{w})$ uses a small language model mDeBERTaV3-base [15]) to map $\mathbf{x}$ into an embedding in $\mathbb{R}^p$, and $T$ learnable LLM embeddings $\{\mathbf{k}_t \in \mathbb{R}^p : t = 1, \ldots, T\}$ for the $T$ LLMs. For a query $\mathbf{x}_i$, the RouterDC generates a selection probability distribution over $T$ LLMs as

$$R(\mathbf{x}_i; \boldsymbol{\theta}) = \text{softmax}\left[\text{sim}(\mathcal{E}(\mathbf{x}_i; \mathbf{w}), \mathbf{k}_1), \ldots, \text{sim}(\mathcal{E}(\mathbf{x}_i; \mathbf{w}), \mathbf{k}_T)\right], \tag{3}$$

where $\boldsymbol{\theta} \equiv \{\mathbf{w}, \mathbf{k}_1, \mathbf{k}_2, \ldots, \mathbf{k}_T\}$ denotes the set of the parameters in RouterDC, $\text{sim}(\cdot, \cdot)$ denotes the cosine similarity, and $\text{softmax}(\cdot)$ denotes the softmax normalization.

One can train the router by minimizing the distance between the output of the router and a score distribution over $\{s_i^{(t)} : t = 1, \ldots, T\}$, i.e., $\min_{\boldsymbol{\theta}} \sum_{(\mathbf{x}_i, y_i) \in \mathcal{D}_{\text{train}}} \text{KL}\left(R(\mathbf{x}_i; \boldsymbol{\theta}), \text{softmax}[s_i^{(1)}, \ldots, s_i^{(T)}]\right),$

**Algorithm 1** Query-Based **Router** by **D**ual **C**ontrastive Learning (**RouterDC**)

---

**Input:** training set $\mathcal{D}_{\text{train}}$, LLMs $\{\mathcal{M}_t : t = 1, \ldots, T\}$, #positive LLMs $K_+$, #negative LLMs $K_-$, #out-group queries $H$, #clusters $N$, hyper-parameter $\lambda$, mini-batch size $b$, and learning rate $\eta$; learnable parameters $\boldsymbol{\theta}$: encoder $\mathcal{E}(\cdot; \mathbf{w})$ and LLM embeddings $\{\mathbf{k}_t : t = 1, \ldots, T\}$;

    *training:*
1:   score LLMs for each sample $(\mathbf{x}_i, y_i) \in \mathcal{D}_{\text{train}}$ and obtain $\{s_i^{(t)} : t = 1, \ldots, T\}$;
2:   cluster training queries $\{\mathbf{x}_i : i = 1, \ldots, n\}$ into $N$ groups;
3:   **repeat**
4:      sample a mini-batch of data $\mathcal{B}$ from $\mathcal{D}_{\text{train}}$;
5:      **for** $(\mathbf{x}_i, y_i) \in \mathcal{B}$ **do**
6:          obtain positive LLMs index set $\mathcal{I}_i^+$ and negative LLMs index set $\mathcal{I}_i^-$;
7:          compute the sample-LLM contrastive loss $\mathcal{L}_{\text{sample-LLM}}(\mathbf{x}_i, y_i; \boldsymbol{\theta})$ by Eq. (4);
8:          sample an in-group query $\mathbf{x}_i^+$ and a set of $H$ out-group queries $\mathcal{X}_i^-$ from $\mathcal{B}$;
9:          compute the sample-sample contrastive loss $\mathcal{L}_{\text{sample-sample}}(\mathbf{x}_i; \boldsymbol{\theta})$ by Eq. (5);
10:     **end for**
11:     $\mathcal{L}(\mathcal{B}; \boldsymbol{\theta}) = \sum_{(\mathbf{x}_i, y_i) \in \mathcal{B}} \mathcal{L}_{\text{sample-LLM}}(\mathbf{x}_i, y_i; \boldsymbol{\theta}) + \lambda \, \mathcal{L}_{\text{sample-sample}}(\mathbf{x}_i; \boldsymbol{\theta})$;
12:     $\boldsymbol{\theta} \leftarrow \boldsymbol{\theta} - \eta \nabla_{\boldsymbol{\theta}} \mathcal{L}(\mathcal{B}; \boldsymbol{\theta})$;
13: **until** converged.

    *inference:*
14: sample a testing query $\mathbf{x}'$;
15: $t' = \arg \max_{t \in \{1, \ldots, T\}} \text{sim}(\mathcal{E}(\mathbf{x}'; \mathbf{w}), \mathbf{k}_t)$;
16: $\hat{\mathbf{y}}' = \mathcal{M}_{t'}(\mathbf{x}')$.

---

where $\mathsf{KL}(\cdot, \cdot)$ is the Kullback-Leibler divergence [29]. This KL loss is recently used in [33] for LLM routing, but we argue that it may not be a good proxy for training the router since the goal of the router is to assign queries to *top-performing* LLMs instead of aligning the scores with $R(\mathbf{x}_i; \boldsymbol{\theta})$, particularly for the *bottom-performing* LLMs.

We draw inspiration from contrastive learning [37, 23] and propose a sample-LLM contrastive loss to learn the router. For a query $\mathbf{x}_i$, we construct its positive LLMs index set $\mathcal{I}_i^+$ and its negative LLMs index set $\mathcal{I}_i^-$ based on the scores $\{s_i^{(t)} : t = 1, \ldots, T\}$ as: $\mathcal{I}_i^+$ consists of the indices of LLMs corresponding to the top-$K_+$ scores, while $\mathcal{I}_i^-$ consists of the indices of LLMs corresponding to the bottom-$K_-$ scores with $s_i^{(t)} < 0.5$. Note that $K_+$ can be larger than 1 ($K_+ = 3$ in our experiments) as there can be multiple LLMs that are suitable for a query in practice. We expect the router to pull the query embedding $\mathcal{E}(\mathbf{x}_i; \mathbf{w})$ closer to the positive LLMs' embeddings $\{\mathbf{k}_{t_+} : t_+ \in \mathcal{I}_i^+\}$ while pushing apart from the negative LLMs' embeddings $\{\mathbf{k}_{t_-} : t_- \in \mathcal{I}_i^-\}$. To this end, we propose the sample-LLM contrastive loss as

$$\mathcal{L}_{\text{sample-LLM}}(\mathbf{x}_i, y_i; \boldsymbol{\theta}) = \sum_{t_+ \in \mathcal{I}_i^+} - \log \frac{e^{\text{sim}(\mathcal{E}(\mathbf{x}_i; \mathbf{w}), \mathbf{k}_{t_+})}}{e^{\text{sim}(\mathcal{E}(\mathbf{x}_i; \mathbf{w}), \mathbf{k}_{t_+})} + \sum_{t_- \in \mathcal{I}_i^-} e^{\text{sim}(\mathcal{E}(\mathbf{x}_i; \mathbf{w}), \mathbf{k}_{t_-})}}. \tag{4}$$

### 3.4 Sample-Sample Contrastive Loss

We empirically find that training the router by minimizing the sample-LLM contrastive loss alone is not stable (refer to Figure 12 in Section 4.4). The reason is that some similar queries can have dissimilar embeddings and may be routed to different LLMs. To improve the robustness of the router, we introduce a sample-sample contrastive loss to encourage the encoder to generate similar embeddings for similar queries.

First, we cluster queries into multiple groups by unsupervised clustering. Specifically, we extract the embeddings of all training queries using a pre-trained encoder (i.e., mDeBERTaV3-base [15]) and transform them into low-dimensional vectors by the t-SNE algorithm [48]. Then the $k$-means clustering algorithm [35] is used to cluster these low-dimensional vectors into $N$ groups $\{\mathcal{K}_1, \ldots, \mathcal{K}_N\}$.

Next, we construct a sample-sample contrastive loss to encourage samples in the same group to have similar embeddings. Specifically, for a query $\mathbf{x}_i \in \mathcal{K}_j$, we randomly select an in-group query $\mathbf{x}_i^+ \in \mathcal{K}_j$ and an out-group set $\mathcal{X}_i^- \subset \{\cup_{j' \neq j} \mathcal{K}_{j'}\}$ of $H$ queries from the training mini-batch at each iteration. Similar to the sample-LLM contrastive loss, we propose a sample-sample contrastive loss to pull the embedding of $\mathbf{x}_i$ closer to the embedding of $\mathbf{x}_i^+$ while pushing it away from the embedding of queries in $\mathcal{X}_i^-$. Formally, the sample-sample contrastive loss is formulated as

$$\mathcal{L}_{\text{sample-sample}}(\mathbf{x}_i; \boldsymbol{\theta}) = -\log \frac{e^{\text{sim}(\mathcal{E}(\mathbf{x}_i; \mathbf{w}), \mathcal{E}(\mathbf{x}_i^+; \mathbf{w}))}}{e^{\text{sim}(\mathcal{E}(\mathbf{x}_i; \mathbf{w}), \mathcal{E}(\mathbf{x}_i^+; \mathbf{w}))} + \sum_{\mathbf{x}_i^- \in \mathcal{X}_i^-} e^{\text{sim}(\mathcal{E}(\mathbf{x}_i; \mathbf{w}), \mathcal{E}(\mathbf{x}_i^-; \mathbf{w}))}}. \quad (5)$$

### 3.5 Training and Inference

**Training.** We learn a router $R(\mathbf{x}; \boldsymbol{\theta})$ by minimizing the final objective consisting of sample-LLM and sample-sample contrastive losses, i.e.,

$$\mathcal{L}(\mathcal{D}_{\text{train}}; \boldsymbol{\theta}) = \sum_{(\mathbf{x}_i, y_i) \in \mathcal{D}_{\text{train}}} \mathcal{L}_{\text{sample-LLM}}(\mathbf{x}_i, y_i; \boldsymbol{\theta}) + \lambda \, \mathcal{L}_{\text{sample-sample}}(\mathbf{x}_i; \boldsymbol{\theta}), \quad (6)$$

where $\lambda > 0$ is a hyper-parameter. In our experiments, $\lambda$ is set to 1.

RouterDC contains fewer than 100M parameters (that is, the encoder model $\mathcal{E}(\mathbf{x}; \mathbf{w})$ is small and the number of parameters in the LLM embeddings $\{\mathbf{k}_1, \ldots, \mathbf{k}_T\}$ is negligible), thus it is parameter-efficient. Moreover, training the router is computationally efficient as it does not require backpropagating the gradients through the LLMs.

**Inference.** During inference, for each testing query $\mathbf{x}'$, we compute $R(\mathbf{x}'; \boldsymbol{\theta})$ and select the LLM with the largest probability, i.e., $t' = \arg\max_{t \in \{1, \ldots, T\}} \text{sim}(\mathcal{E}(\mathbf{x}'; \mathbf{w}), \mathbf{k}_t)$. Then we generate the prediction as $\hat{\mathbf{y}}' = \mathcal{M}_{t'}(\mathbf{x}')$.

Compared with existing LLM assembling methods like voting [31] and cascade [2], which may call LLMs multiple times for a query, RouterDC is much more efficient as it only needs to call the selected LLM once.

## 4 Experiments

### 4.1 Experimental Setup

**Candidate LLMs.** We choose seven open-source LLMs from HuggingFace[1]: (i) *Mistral-7B* [24] is a general LLM released by the Mistral-AI team; (ii) *MetaMath-Mistral-7B* [54] is fine-tuned on the MetaMathQA dataset [54]; (iii) *zephyr-7b-beta* [47] is an aligned version of Mistral-7B using direct preference optimization [40] on a mix of publicly available, synthetic datasets; (iv) *Chinese-Mistral-7B* [58] expands the vocabulary and incrementally pre-trains Mistral-7B on Chinese corpus; (v) *dolphin-2.6-mistral-7b* [8] is fine-tuned from Mistral-7B and released by Cognitive Computations; (vi) *Llama-3-8B* [45] is a general LLM developed by the Meta company; (vii) *dolphin-2.9-llama3-8b* [9] is fine-tuned from Llama-3-8B and released by Cognitive Computations. The first five LLMs are Mistral-based, while the last two LLMs are Llama-3-based.

**Datasets.** We evaluate in various tasks: (i) MMLU [17] is a general benchmark that covers 57 subjects; (ii) GSM8K [7] is a mathematical benchmark with diverse grade school questions; (iii) CMMLU [30] is a comprehensive Chinese benchmark that covers 67 subjects ranging from basic to advanced professional levels; (iv) ARC-C [6] is a reasoning benchmark containing different grade-school level questions; and (v) HumanEval [3] is a code completion benchmark consisting of programming problems assessing language comprehension, algorithms, and simple mathematics. For GSM8K, we use its default training and testing split. As the rest tasks do not have a default split, we randomly split the dataset into training (70%) and testing (30%) sets. All the training sets are unioned together to form the total training set $\mathcal{D}_{\text{train}}$ for learning the router. The learned router is then evaluated on the testing set of *in-distribution* tasks.

We also evaluate the trained router on three *out-of-distribution* (OOD) tasks: (i) PreAlgebra [18], which consists of basic university-level algebra problems; (ii) MBPP [1], which is a code benchmark

---

[1] https://huggingface.co/

Table 1: Testing accuracy (%) on in-distribution tasks. "Time" denotes the total inference time in minutes. The best is in **bold** and the second-best is underlined.

| | | MMLU | GSM8K | CMMLU | ARC-C | HumanEval | Avg | Time (m) |
|---|---|---|---|---|---|---|---|---|
| *Candidate LLMs* | Mistral-7B [24] | 62.14 | 36.71 | 43.83 | 49.43 | 28.98 | 44.22 | 6.94 |
| | MetaMath-Mistral-7B [54] | 59.86 | 69.63 | 43.83 | 48.30 | 29.80 | 50.28 | 7.23 |
| | zephyr-7b-beta [47] | 59.81 | 33.00 | 42.82 | 57.95 | 22.04 | 43.13 | 6.73 |
| | Chinese-Mistral-7B [58] | 57.42 | 41.03 | 49.67 | 43.47 | 21.43 | 42.60 | 7.11 |
| | dolphin-2.6-mistral-7b [8] | 60.53 | 52.38 | 43.71 | 52.56 | 45.10 | 50.86 | 6.91 |
| | Meta-Llama-3-8B [45] | **64.59** | 47.76 | **51.77** | 49.43 | 26.73 | 48.06 | 6.33 |
| | dolphin-2.9-llama3-8b [9] | 59.46 | 69.81 | 44.72 | 49.43 | 49.39 | 54.56 | 5.33 |
| | Voting [31] | 63.30 | 67.39 | 47.48 | 50.85 | 42.85 | 54.37 | 46.59 |
| *Routing* | CosineClassifier | 59.72 | 69.03 | 45.47 | 50.57 | 46.33 | 54.22 | 8.30 |
| | ZOOTER [33] | 60.48 | 66.69 | 45.27 | 53.13 | 44.29 | 53.97 | 8.01 |
| | LoraRetriever (clustering) [56] | 63.33 | 66.63 | **51.77** | 57.10 | 40.00 | 55.77 | 7.86 |
| | RouterDC | 61.07 | **70.32** | **51.77** | **58.52** | **51.02** | **58.54** | 7.97 |

Table 2: Testing accuracy (%) on out-of-distribution tasks. "Time" denotes the total inference time in minutes. The best is in **bold** and the second-best is underlined.

| | | PreAlgebra | MBPP | C-EVAL | Avg | Time (m) |
|---|---|---|---|---|---|---|
| *Candidate LLMs* | Mistral-7B [24] | 24.80 | 37.90 | 46.43 | 36.38 | 4.31 |
| | MetaMath-Mistral-7B [54] | 39.15 | 37.74 | 45.17 | 40.69 | 4.13 |
| | zephyr-7b-beta [47] | 20.78 | 31.14 | 44.87 | 32.26 | 4.30 |
| | Chinese-Mistral-7B [58] | 18.48 | 29.64 | 48.44 | 32.19 | 4.40 |
| | dolphin-2.6-mistral-7b [8] | 29.28 | 44.86 | 45.10 | 39.75 | 3.20 |
| | Meta-Llama-3-8B [45] | 27.67 | 43.02 | **52.01** | 40.90 | 3.95 |
| | dolphin-2.9-llama3-8b [9] | **39.72** | **47.34** | 44.80 | 43.95 | 3.15 |
| | Voting [31] | 39.03 | 41.60 | 48.50 | 43.04 | 27.43 |
| *Routing* | CosineClassifier | 36.97 | 38.48 | 47.77 | 41.07 | 4.43 |
| | ZOOTER [33] | 34.44 | 41.10 | 44.95 | 40.16 | 4.28 |
| | LoraRetriever (clustering) [56] | 35.36 | 43.12 | **52.01** | 43.50 | 4.22 |
| | RouterDC | 38.81 | 46.80 | 51.93 | **45.85** | 4.24 |

that consists of 1,000 crowd-sourced Python programming problems; and (iii) C-EVAL [21], which is a comprehensive Chinese evaluation benchmark spanning 52 diverse disciplines and four difficulty levels.

**Baselines.** We compare RouterDC with the following baselines: (i) CosineClassifier, which treats the routing problem as a multi-class classification (the top-1 LLM is the label) and trains a cosine classifier on outputs of the encoder. CosineClassifier is equivalent to RouterDC with $K_+ = 1$, $K_- = T - 1$, and $\lambda = 0$; (ii) Voting [31], which feeds the query to all LLMs and chooses the final prediction by majority voting; (iii) ZOOTER [33], which trains a router by supervised learning using rewards obtained by the scoring method in Section 3.2; (iv) LoraRetriever [56] designs a routing strategy based on task identities, which are unavailable in practice and we replace them with the cluster indices obtained by the clustering method in Section 3.4.

**Implementation Details.** Following [5], we use the Language Model Evaluation Harness package [11] for evaluation. For open-ended generation questions, we query LLMs $M = 10$ times by employing beam sampling with a temperature of $0.2$ to calculate the score. For the router, we adopt mDeBERTaV3-base [16] as the encoder $\mathcal{E}(\mathbf{x}; \mathbf{w})$, which is a small language model containing only 86M parameters. The dimension of each LLM embedding is set to 768. The hyper-parameters $K_+, K_-, H$, and $\lambda$ are set to $3, 3, 3$, and $1$, respectively. The number of clusters $N$ is set to 5. The router is trained for 1000 steps using the AdamW [32] optimizer with a learning rate of $5 \times 10^{-5}$, a weight decay of $0.01$, and a mini-batch size of $64$. All experiments are run on NVIDIA A100 80GB GPUs.

### 4.2 Main Results

**In-Distribution Results.** Table 1 shows the testing accuracies on five in-distribution tasks. As can be seen, RouterDC achieves the highest average accuracy, surpassing the best individual LLM (i.e., dolphin-2.9-llama3-8b) by a large margin of 3.98%. RouterDC achieves accuracy improvements over the top-performing individual model on three tasks, with an increase of $+0.51\%$ for GSM8K, $+0.57\%$

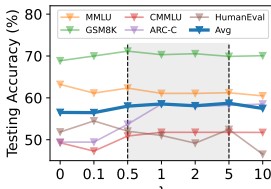
Figure 5: Effects of $\lambda$.

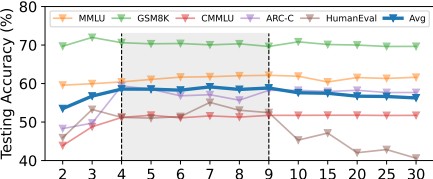
Figure 6: Effects of #clusters $N$.

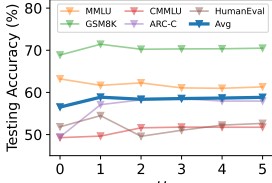
Figure 7: Effects of $H$.

for ARC-C, and $+1.63\%$ for HumanEval. Compared with ZOOTER and CosineClassifier, RouterDC consistently performs better on all the tasks, demonstrating that the proposed dual contrastive losses can train a more effective router. Furthermore, RouterDC achieves an average accuracy improvement of $+2.77\%$ over LoraRetriever, validating the usefulness of the sample-LLM contrastive loss. Additionally, RouterDC, with only $28.3$ minutes for training, outperforms Voting on four of five tasks and is about $6\times$ faster in inference.

**Out-of-Distribution Results.** Table 2 shows the testing accuracy on three OOD tasks. As can be seen, the proposed RouterDC again achieves the highest accuracy on average, exceeding the best-performing individual LLM (i.e., *dolphin-2.9-llama3-8b*) by a large margin of $1.9\%$. For each task, RouterDC has roughly comparable performance with the best-performing individual LLM, e.g., $38.81$ vs. $39.72$ on PreAlgebra, $46.80$ vs. $47.34$ on MBPP, and $51.93$ vs. $52.01$ on C-EVAL, which demonstrates that RouterDC can select suitable LLMs for queries from OOD tasks. Among all routing methods, only our RouterDC can surpass *dolphin-2.9-llama3-8b*, confirming that RouterDC has a better generalization ability. Compared with Voting, RouterDC performs better on all tasks except PreAlgebra, on which they are comparable.

### 4.3 Sensitivity Analysis

**Effects of $\lambda$.** We conduct an experiment to study the effect of $\lambda$ in Eq. (6) w.r.t. the testing accuracy. From Figure 5 (the detailed results are in Table 5 of Appendix A), we can see that using two contrastive losses together (i.e., $\lambda = 1$) achieves better overall performance than using the sample-LLM contrastive loss alone (i.e., $\lambda = 0$). Moreover, the overall performance of RouterDC is insensitive to a wide range of $\lambda \in [0.5, 5]$, making it easy to choose the value of $\lambda$ in practice.

**Effects of number of clusters $N$.** We conduct an experiment to study the effect of the number of clusters (i.e., $N$) used in the sample-sample contrastive loss w.r.t. the testing accuracy. From Figure 6, we can find that RouterDC is insensitive to a wide range of $N \in [4, 9]$. Moreover, increasing $N$ leads to higher average accuracy when $N$ is small ($\leq 4$), but the accuracy saturates quickly.

**Effects of number of out-group queries $H$.** Figure 7 shows the testing accuracy with $H$. When $= 0$, $\mathcal{L}_{\text{sample-sample}}$ is constant, which means using $\mathcal{L}_{\text{sample-LLM}}$ alone is not the best configuration. Moreover, the values of $H \geq 1$ play a negligible influence on the average performance of RouterDC.

**Effects of $K_+$ and $K_-$.** To investigate the sensitivity of $K_+$ and $K_-$, we conduct an experiment using the setting in Section 4.1. Figure 8 shows the average testing accuracy w.r.t. $K_+$ and $K_-$ with the in-distribution setting. As can be seen, for all the configurations, RouterDC outperforms the best individual LLM (i.e., $54.56\%$ for *dolphin-2.9-llama3-8b* in Table 1). Note that among all the configurations, RouterDC (with $K_+ = 1$ and $K_- = 6$) performs worse, showing that selecting only the top-1 LLM as positive and the other LLMs as negative is inappropriate for learning the router.

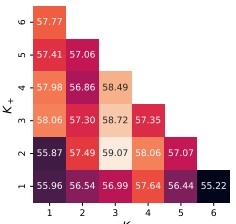
Figure 8: Average testing accuracy w.r.t. $K_+$ and $K_-$ on five in-distribution tasks. Lighter color indicates higher accuracy.

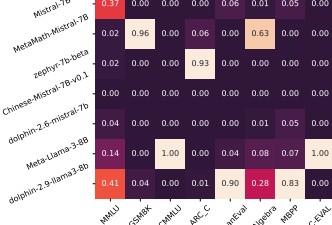
Figure 9: Distribution of testing queries over LLMs. Lighter color indicates higher percentage.

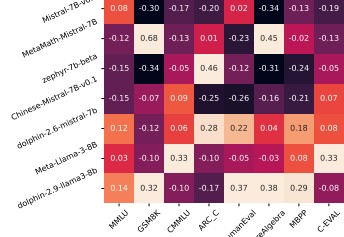
Figure 10: Average cosine similarity between LLMs and query embeddings. Lighter color indicates higher similarity.

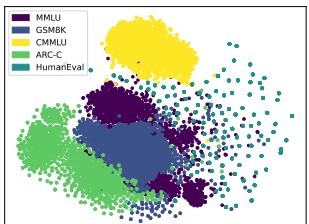

Figure 11: Visualization of embeddings of training queries.

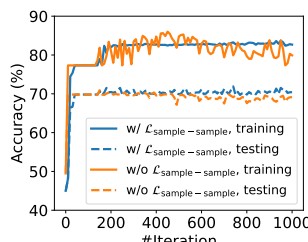

Figure 12: Training and testing accuracy curves of RouterDC on GSM8K.

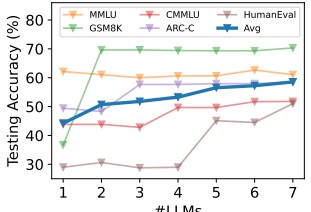

Figure 13: Testing accuracy with different numbers of LLMs.

Table 3: Robustness of RouterDC to LLM losses during inference.

|  | MMLU | GSM8K | CMMLU | ARC-C | HumanEval | Avg |
|---|---|---|---|---|---|---|
| All | 61.07 | 70.32 | 51.77 | 58.52 | 51.02 | 58.54 |
| w/o Mistral-7B | 62.26 | 70.32 | 51.77 | 58.52 | 50.41 | 58.66 |
| w/o MetaMath-Mistral-7B | 60.93 | 69.81 | 51.77 | 57.67 | 51.02 | 58.24 |
| w/o zephyr-7b-beta | 61.19 | 70.32 | 51.77 | 53.13 | 51.02 | 57.49 |
| w/o Chinese-Mistral-7B | 61.07 | 70.32 | 51.77 | 58.52 | 51.02 | 58.54 |
| w/o dolphin-2.6-mistral-7b | 60.95 | 70.32 | 51.77 | 58.52 | 51.02 | 58.52 |
| w/o meta-llama/Meta-Llama-3-8B | 60.95 | 70.32 | 46.30 | 58.52 | 51.02 | 57.42 |
| w/o dolphin-2.9-llama3-8b | 61.36 | 69.41 | 51.74 | 57.95 | 46.53 | 57.40 |

## 4.4 Analysis

**Does RouterDC select the suitable LLM for each query?** To answer this question, we analyze the assignment of testing queries to LLMs in each task. Figure 9 shows the distribution, which has a clear structure on both in-distribution and out-distribution tasks. For example, most GSM8K and PreAlgebra queries are assigned to MetaMath-Mistral-7B and dolphin-2.9-llama3-8b, which have strong mathematical ability (Tables 1 and 2). To further investigate the routing rule of RouterDC, we compute the average cosine similarity between LLMs and the query embeddings for each task. As shown in Figure 10, the similarity matrix is roughly aligned with the assignment matrix in Figure 9. For example, embeddings of GSM8K and PreAlgrebra queries are more similar to MetaMath-Mistral-7B and dolphin-2.9-llama3-8b than to other LLMs.

**Visualization of Training Queries.** Figure 11 shows the t-SNE visualization [48] of the embeddings of training queries using a pre-trained encoder mDeBERTaV3-base [15]. As shown, except for HumanEval, all tasks have a clear clustering structure, confirming that using unsupervised clustering in Section 3.4 is reasonable.

**Effectiveness of $\mathcal{L}_{\text{sample-sample}}$.** We conduct experiments to study the effectiveness of $\mathcal{L}_{\text{sample-sample}}$ (Eq. (5)). Figure 12 shows the training and testing accuracy curves of RouterDC (with or without $\mathcal{L}_{\text{sample-sample}}$) on GSM8K. As can be seen, the training curve of RouterDC (w/o $\mathcal{L}_{\text{sample-sample}}$) exhibits considerable oscillation, whereas that of RouterDC is much more stable. Figure 15(a) in Appendix B shows t-SNE visualization of training query embeddings extracted by the trained encoder of RouterDC (w/o $\mathcal{L}_{\text{sample-sample}}$). As can be seen, query embeddings belonging to different tasks are roughly mixed together. Example 2 in Appendix B provides two similar GSM8K queries, which both require basic calculation of shopping costs. Their embeddings have very low similarity (only $-0.4589$) when the router is trained by $L_{\text{sample-LLM}}$ alone. After integrating $L_{\text{sample-sample}}$, training query embeddings have a clear cluster structure (Figure 15(b)) with the similarity between these two example queries increases to $0.9982$. Furthermore, RouterDC achieves higher testing accuracy than its counterpart, verifying the effectiveness of $\mathcal{L}_{\text{sample-sample}}$.

**Routing to Different Numbers of LLMs.** We evaluate the performance of RouterDC when the number of LLMs increases. Figure 13 shows the testing accuracies on five in-distribution tasks. As can be seen, adding LLMs consistently enhances the average accuracy. Table 8 in Appendix A shows the detailed results and configurations.

**Robustness to LLM Losses during Inference.** In a production environment, the loss of model servers is sometimes unavoidable due to various reasons such as network problems, thus placing crucial requirements on the robustness of the router. We conduct an experiment to validate the

robustness of RouterDC by removing an LLM during inference. Table 3 shows the testing accuracies on five in-distribution tasks. We can see that RouterDC reliably withstands the loss of any single LLM. The robustness is attributed to the fact that multiple LLMs (with top scores) are chosen as positive labels in the sample-LLM contrastive loss, and they can be regarded as each other's backup.

**Cost-Effectiveness.** As cost is an important metric to evaluate LLMs, following [19], we conduct experiments on two tasks (i.e., GSM8K and MBPP) to consider the LLM costs. We modify the score $s_i^{(t)}$ to $s_i^{(t)} + c_i^{(t)}$, where $c_i^{(t)}$ is the cost of query $\mathbf{x}_i$ using the $t$th LLM. As can be seen from Figure 14, RouterDC is more cost-effective than CosineClassifier and ZOOTER on both tasks.

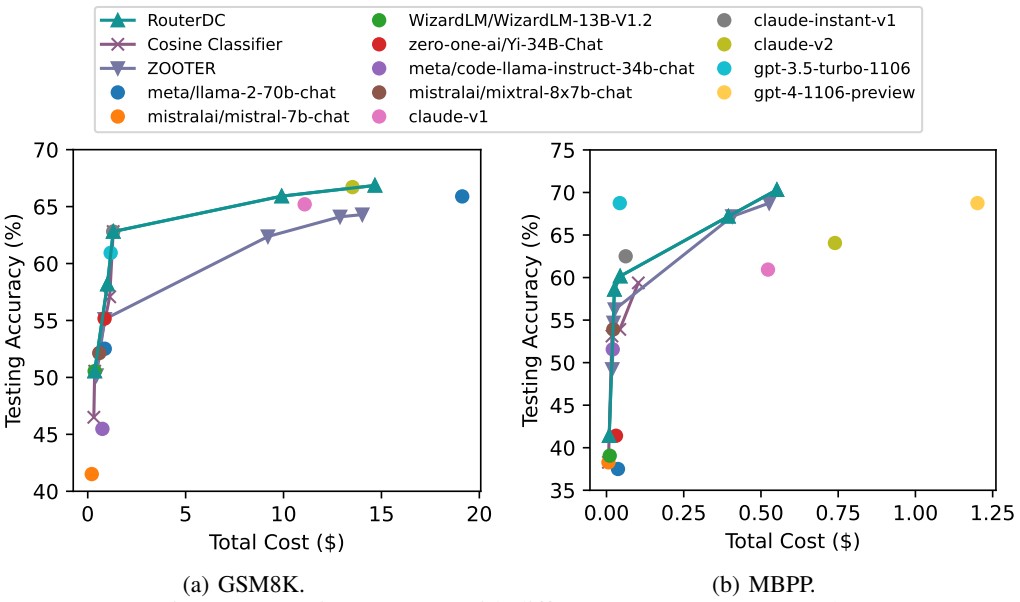

(a) GSM8K.  (b) MBPP.

Figure 14: Testing accuracy with different costs on RouterBench.

**Discussions on Availability of Task Identity.** In Section 3.4, we cluster samples into $N$ groups and apply the sample-sample contrastive training to encourage similar queries with similar embeddings. However, when the task identity is available in the training dataset, the samples can be naturally grouped into different tasks. To explore the performance of RouterDC with additional task identity, we replace the $\{\mathcal{K}_1, \ldots, \mathcal{K}_N\}$ with the groups of different tasks and conduct experiments on five in-distribution tasks. Table 4 shows the testing accuracy comparison between RouterDC and its variant. As can be seen, RouterDC is comparable to RouterDC (w/ task identity), showing the effectiveness of the unsupervised clustering.

Table 4: Testing accuracy(%) of RouterDC w/ or w/o task identity.

|  | MMLU | GSM8K | CMMLU | ARC-C | HumanEval | Avg |
|---|---|---|---|---|---|---|
| RouterDC (w/o task identity) | 61.07 | 70.32 | 51.77 | 58.52 | 51.02 | 58.54 |
| RouterDC (w/ task identity) | 64.49 | 69.63 | 51.77 | 57.95 | 49.39 | 58.65 |

## 5 Conclusion

In this paper, we study the problem of training a router to assemble LLMs. We propose RouterDC to learn a query-based router using two novel contrastive losses (i.e., the sample-LLM and sample-sample contrastive losses). Experimental results show that RouterDC effectively assembles LLMs and outperforms individual top-performing LLMs as well as existing routing methods on both in-distribution and out-distribution tasks. As the proposed two contrastive losses are general, we consider applying them to other routing problems in future work.

## Acknowledgements

This work is supported by NSFC key grant under grant no. 62136005, NSFC general grant under grant no. 62076118, Shenzhen fundamental research program JCYJ20210324105000003, and the Research Grants Council of the Hong Kong Special Administrative Region (Grants C7004-22G-1 and 16202523).

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

# A  Full Results of Figures 5, 6, 7, and 13

Table 5 is the full results of Figure 5 (i.e., the testing accuracy with different $\lambda$'s). As can be seen, RouterDC is insensitive to a wide range of $\lambda \in [0.5, 5]$.

Table 5: Testing accuracy (%) with different $\lambda$'s.

| $\lambda$ | MMLU | GSM8K | CMMLU | ARC-C | HumanEval | Avg |
|---|---|---|---|---|---|---|
| 0 | 63.21 | 68.87 | 49.27 | 49.43 | 51.84 | 56.52 |
| 0.1 | 61.14 | 70.00 | 47.28 | 49.43 | 54.49 | 56.47 |
| 0.2 | 60.41 | 69.20 | 47.14 | 54.83 | 52.45 | 56.81 |
| 0.5 | 62.33 | 71.16 | 50.88 | 53.69 | 52.04 | 58.02 |
| 1 | 61.07 | 70.32 | 51.77 | 58.52 | 51.02 | 58.54 |
| 2 | 61.07 | 70.55 | 51.77 | 57.95 | 49.18 | 58.11 |
| 5 | 61.22 | 69.91 | 51.77 | 58.24 | 52.45 | 58.72 |
| 10 | 60.48 | 70.07 | 51.74 | 58.52 | 46.53 | 57.47 |

Table 6 is the full results of Figure 6 (i.e., the testing accuracy with different #clusters $N$'s). We can see that RouterDC is insensitive to a wide range of $N \in [4, 9]$.

Table 6: Testing accuracy (%) with different $N$'s.

| #Clusters | MMLU | GSM8K | CMMLU | ARC-C | HumanEval | Avg |
|---|---|---|---|---|---|---|
| 2 | 59.58 | 69.70 | 43.88 | 48.30 | 45.92 | 53.48 |
| 3 | 59.96 | 71.98 | 48.78 | 49.72 | 53.27 | 56.74 |
| 4 | 60.43 | 70.61 | 51.19 | 59.37 | 51.22 | 58.56 |
| 5 | 61.07 | 70.32 | 51.77 | 58.52 | 51.02 | 58.54 |
| 6 | 61.67 | 70.40 | 51.10 | 56.82 | 51.42 | 58.28 |
| 7 | 61.78 | 70.05 | 51.60 | 57.10 | 55.10 | 59.13 |
| 8 | 62.02 | 70.32 | 51.28 | 55.68 | 53.06 | 58.47 |
| 9 | 62.14 | 69.63 | 51.74 | 58.24 | 52.45 | 58.84 |
| 10 | 61.90 | 70.84 | 51.74 | 58.24 | 45.31 | 57.61 |
| 15 | 60.41 | 70.14 | 51.77 | 57.95 | 47.14 | 57.48 |
| 20 | 61.55 | 70.00 | 51.77 | 58.24 | 42.04 | 56.72 |
| 25 | 61.33 | 69.63 | 51.71 | 57.67 | 42.85 | 56.64 |
| 30 | 61.62 | 69.65 | 51.74 | 57.67 | 40.61 | 56.26 |

Table 7 is the full results of Figure 7 (i.e., the testing accuracy with different #out-group queries $H$'s). As we can see, RouterDC is robust across various $H$'s, except for $H = 0$, which is equivalent to using $\mathcal{L}_{\text{sample-sample}}$ alone.

Table 7: Testing accuracy (%) with different $H$'s.

| $H$ | MMLU | GSM8K | CMMLU | ARC-C | HumanEval | Avg |
|---|---|---|---|---|---|---|
| 0 | 63.21 | 68.87 | 49.27 | 49.43 | 51.84 | 56.52 |
| 1 | 61.67 | 71.43 | 49.64 | 57.10 | 54.49 | 58.87 |
| 2 | 62.26 | 70.24 | 51.60 | 58.24 | 49.59 | 58.38 |
| 3 | 61.07 | 70.32 | 51.77 | 58.52 | 51.02 | 58.54 |
| 4 | 60.98 | 70.36 | 51.74 | 57.95 | 52.24 | 58.66 |
| 5 | 61.31 | 70.49 | 51.74 | 57.95 | 52.65 | 58.83 |

Table 8 is the full results of Figure 13 (i.e., the testing accuracy with #LLMs). As can be seen, adding LLMs consistency enhances the average accuracy.

Table 8: Testing accuracy (%) with #LLMs.

| | #LLMs | MMLU | GSM8K | CMMLU | ARC-C | HumanEval | Avg |
|---|---|---|---|---|---|---|---|
| Mistral-7B | 1 | 62.14 | 36.71 | 43.83 | 49.43 | 28.98 | 44.22 |
| +MetaMath-Mistral-7B | 2 | 61.07 | 69.63 | 43.83 | 48.30 | 30.62 | 50.69 |
| +zephyr-7b-beta | 3 | 59.98 | 69.63 | 42.82 | 57.67 | 28.78 | 51.78 |
| +Chinese-Mistral-7B | 4 | 60.63 | 69.42 | 49.67 | 57.67 | 28.98 | 53.27 |
| +dolphin-2.6-mistral-7b | 5 | 60.65 | 69.33 | 49.67 | 57.95 | 45.10 | 56.54 |
| +meta-llama/Meta-Llama-3-8B | 6 | 62.64 | 69.34 | 51.71 | 57.95 | 44.49 | 57.23 |
| +dolphin-2.9-llama3-8b | 7 | 61.07 | 70.32 | 51.77 | 58.52 | 51.02 | 58.54 |

# B    Detailed Results for Section 4.4

**Example 2**

**Query 1:** Mary does her grocery shopping on Saturday. She does her shopping only at a specific store where she is allowed a credit of $100, which must be paid in full before her next shopping trip. That week she spent the full credit limit and paid $15 of it on Tuesday and $23 of it on Thursday. How much credit will Mary need to pay before her next shopping trip?

**Query 2:** Betty is saving money for a new wallet which costs $100. Betty has only half of the money she needs. Her parents decided to give her $15 for that purpose, and her grandparents twice as much as her parents. How much more money does Betty need to buy the wallet?

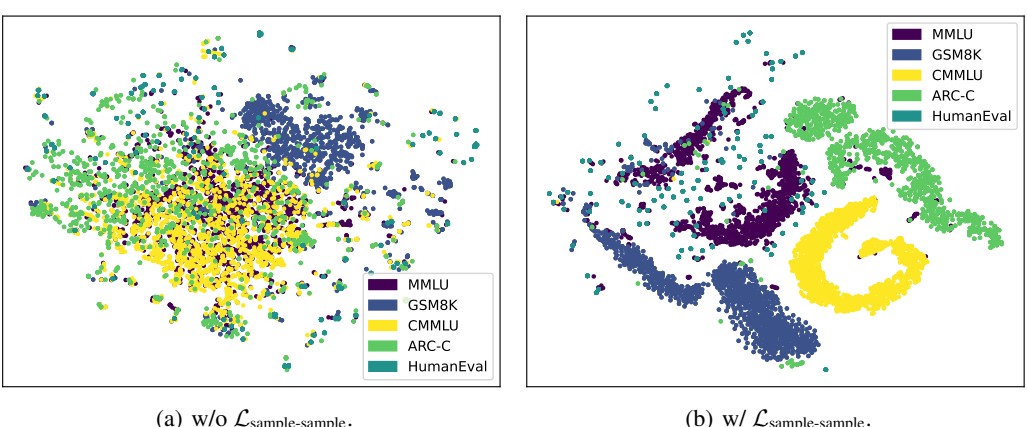

(a) w/o $\mathcal{L}_{\text{sample-sample}}$.  (b) w/ $\mathcal{L}_{\text{sample-sample}}$.

Figure 15: t-SNE visualization of training query embeddings extracted by the learned encoder.

# C    Effectiveness of #training samples

To further investigate the sensitivity of the number of training samples used RouterDC, we conduct an experiment to study the performance of RouterDC with different numbers of training samples per task. As can be seen from Figure 16, the testing accuracy saturates quickly, indicating that a small number of samples is sufficient for learning the router (e.g., 100 samples per task). Moreover, with only 30 samples per task, RouterDC already outperforms the previous SOTA overall (57.21 vs 55.77), demonstrating that our RouterDC does not require a large amount of labeled data for training.

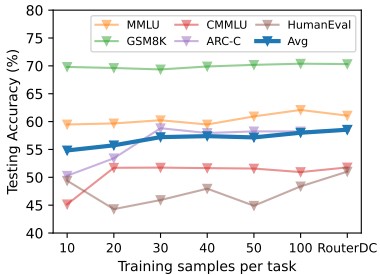

Figure 16: Testing accuracy with different numbers of training samples.

## D  Discussions on Single Task Setting

To validate that RouterDC is a queried-based router rather than a task-wise router, we conduct an experiment in a single task setting, i.e., we train the router on the training set of HumanEval and evaluate it on the testing set. The single task setting is an edge case where all queries may contain the same task information. Hence, the router needs to learn how to route queries appropriately based on the query itself instead of some possible task information contained in the query. Table 9 reports the testing accuracy. As can be seen, RouterDC largely outperforms the best candidate LLM (i.e., dolphin-2.9-llama3-8b) and existing routing methods, demonstrating that the router can select appropriate LLMs for queries based on query characteristics.

Table 9: Testing accuracy (%) on HumanEval task. The best is in **bold**.

| Method | HumanEval |
|---|---|
| Mistral-7B | 28.98 |
| MetaMath-Mistral-7B | 29.80 |
| zephyr-7b-beta | 22.04 |
| Chinese-Mistral-7B | 21.43 |
| dolphin-2.6-mistral-7b | 45.10 |
| Meta-Llama-3-8B | 26.73 |
| dolphin-2.9-llama3-8b | 49.39 |
| ZOOTER | 39.38 |
| CosineClassifier | 52.45 |
| RouterDC | **56.32** |

## E  Discussions on the Distant OOD Task

To further explore the generalization ability of RouterDC, we evaluate the learned router on one more OOD task: JavaScript [57], which aims to generate JavaScript code to solve problems. Different from HumanEval, which generates Python code to solve problems, JavaScript can be viewed as a distant OOD task. Table 10 reports the testing accuracy. As can be seen, RouterDC outperforms existing routing methods by a large margin, demonstrating that our RouterDC is more effective in routing queries of the distant OOD task.

Table 10: Testing accuracy (%) on JavaScript task. The best is in **bold**.

| | JavaScript |
|---|---|
| Mistral-7B | 29.88 |
| MetaMath-Mistral-7B | 31.83 |
| zephyr-7b-beta | 11.71 |
| Chinese-Mistral-7B | 17.68 |
| dolphin-2.6-mistral-7b | 45.00 |
| Meta-Llama-3-8B | 37.07 |
| dolphin-2.9-llama3-8b | 53.84 |
| ZOOTER | 41.64 |
| CosineClassifier | 37.32 |
| RouterDC | **48.66** |

## F  Effectiveness of $L_{\text{sample-sample}}$ for ZOOTER

We conduct experiments to study whether the proposed sample-sample contrastive loss is useful for ZOOTER. Table 11 and Table 12 shows the testing accuracy for the ID and OOD scenarios. As can be seen, integrating $L_{\text{sample-sample}}$ into ZOOTER leads to improvements of $+1.52\%$ and $+0.81\%$ for ID and OOD, respectively, demonstrating that the proposed sample-sample contrastive loss is beneficial for ZOOTER.

Table 11: Testing accuracy (%) of ZOOTER w/ $\mathcal{L}_{\text{sample-sample}}$ on in-distribution tasks.

| | MMLU | GSM8K | CMMLU | ARC-C | HumanEval | Avg |
|---|---|---|---|---|---|---|
| ZOOTER | 60.48 | 66.69 | 45.27 | 53.13 | 44.29 | 53.97 |
| ZOOTER (w/ $\mathcal{L}_{\text{sample-sample}}$) | 60.15 | 69.71 | 46.59 | 54.26 | 46.73 | **55.49** (+1.52) |

Table 12: Testing accuracy (%) of ZOOTER w/ $\mathcal{L}_{\text{sample-sample}}$ on out-of-distribution tasks.

| | Pre-Algebra | MBPP | C-EVAL | Avg |
|---|---|---|---|---|
| ZOOTER | 34.44 | 41.10 | 44.95 | 40.16 |
| ZOOTER (w/ $\mathcal{L}_{\text{sample-sample}}$) | 36.05 | 39.84 | 47.03 | **40.97** (+0.81) |

# G Effectiveness of punishing $s_i^{(t)}$

As mentioned in Section 3.2, we set $s_i^{(t)} = 0$ when the LLM $\mathcal{M}_t$ generates a wrong option for the multiple-choice question $\mathbf{x}_i$. We perform an experiment to verify the effectiveness of such a design. Table 13 shows the testing accuracy on five in-distribution tasks. As can be seen, punishing $s_i^{(t)}$ performs better on average.

Table 13: Testing accuracy (%) of RouterDC with or without setting $s_i^{(t)}$ to 0 for incorrect LLMs.

| | MMLU | GSM8K | CMMLU | ARC-C | HumanEval | Avg |
|---|---|---|---|---|---|---|
| w/o punishing $s_i^{(t)}$ | 61.05 | 70.32 | 49.67 | 56.53 | 52.45 | 58.00 |
| w/ punishing $s_i^{(t)}$ | 61.07 | 70.32 | 51.77 | 58.52 | 51.02 | **58.54** |

# H Limitations

Due to the limited computational resources, we only evaluate RouterDC with candidate LLMs that have relatively small numbers of parameters (i.e., 8B for LLaMA-based LLMs and 7B for Mistral-based LLMs). However, there are many LLMs with more parameters and stronger capabilities available for public use (e.g., LLaMA-2-70B [46] and Mistral-8x7B [24]), making it reasonable to apply the RouterDC to these more capable but expensive models.

Moreover, though RouterDC is designed as a query-based router, the framework can be extended to the chat context, e.g., selecting LLMs based on the recent conversation.

We leave the investigation of such scenarios to future work.

