# OpenReview forum: "RouterDC: Query-Based Router by Dual Contrastive Learning for Assembling Large Language Models"
_NeurIPS.cc/2024/Conference — NeurIPS 2024 poster_

### Official Review · Reviewer_4ok1 · 2024-07-09

**Soundness:** 3
**Presentation:** 4
**Contribution:** 3
**Rating:** 7
**Confidence:** 4

**Summary:**

The paper introduces RouterDC, a novel LLM query router that employs contrastive learning losses to train an encoder and LLM embeddings for routing queries efficiently. The motivation behind this approach is the variability in LLM performance across tasks and domains, and the computational efficiency of routing over ensemble methods. The proposed methodology involves using a small LLM encoder, `mDeBERTaV3-base`, and learnable LLM embeddings for candidate LLMs. RouterDC employs a supervised learning approach with ground truth annotations, using contrastive losses to optimize the routing mechanism. Experimental results demonstrate that RouterDC outperforms existing routing methods on both in-distribution and out-of-distribution tasks.

**Strengths:**

- The paper addresses a significant challenge in LLM utilization, focusing on efficient routing to optimize performance across different tasks and domains.
- The contrastive learning approach for query routing is innovative, leveraging both *sample-LLM contrastive loss* and *sample-sample contrastive loss* to enhance training stability and performance.
- Experimental results show that RouterDC significantly outperforms existing routing methods, indicating the effectiveness of the proposed approach.

**Weaknesses:**

- The parameter efficiency of RouterDC compared to existing routing methods is not addressed, which is important for understanding the scalability of the approach. For example, the training cost is likely significant when the number of LLMs scales up (each training query needs to be evaluated on each LLM), and retraining is required when any new LLMs are incorporated. This could impact the practicality of the system.
- The paper does not incorporate LLM costs into the loss function. When multiple LLMs perform well (which is the motivation of the work), it would be natural to choose the cheapest one to enhance cost efficiency.
- Table 3 shows that removing certain LLMs, such as `Chinese-Mistral-7B` or `dolphin-2.6-mistral-7b`, does not affect overall performance, and removing `Mistral-7B` even improves it. This indicates that including incapable LLMs can lead to unnecessary computational overhead and performance degradation. There is a lack of analysis for this issue, which can be crucial for an effective routing system in the real world.

**Questions:**

1. Table 1 lacks a comparison between different routing methods’ training time or training computational cost. How does RouterDC compare in terms of parameter efficiency with existing methods? This information is crucial for understanding the scalability of the approach.
2. The performance improvements after incorporating sample-sample contrastive loss seem marginal. Can this be further analyzed or explained?
3. On OOD tasks, RouterDC performs worse than the best-performing LLM on certain individual tasks. Is there any analysis of the reasons?
4. Evaluation results on specialized routing benchmarks like [1] could provide a more comprehensive assessment of RouterDC's performance.
5. A normalized average score, instead of the simple average among the scores of different benchmarks, can provide a better comparison in Tables 1 and 2, considering the varying complexity and score scales across benchmarks.
6. There is no need to include RandomSelect as a weak baseline.
7. In the "Routing to Different Numbers of LLMs" evaluation, why were LLMs added in the chosen order? Based on Table 1, adding them in performance-descending order might yield smaller accuracy enhancements.
8. Based on Table 3, the incorporation of certain incapable LLMs can lead to unnecessary computational overhead and performance degradation. Would an LLM-dropping mechanism improve overall performance?
9. The visualization of query embeddings from different benchmarks is confusing. Did you cluster queries within one benchmark or across several benchmarks in the sample-sample contrastive learning loss function?

---
[1] Hu, Qitian Jason, Jacob Bieker, Xiuyu Li, Nan Jiang, Benjamin Keigwin, Gaurav Ranganath, Kurt Keutzer, and Shriyash Kaustubh Upadhyay. "ROUTERBENCH: A Benchmark for Multi-LLM Routing System." arXiv preprint arXiv:2403.12031 (2024).

**Limitations:**

The broader implications of this routing approach, including its scalability and cost efficiency in practical applications, should be more adequately addressed.

---

> ### Author Rebuttal · Authors · 2024-08-07
>
> We sincerely thank you for the detailed and positive comments.
> We carefully address your concerns below.
> Please let us know if you have any follow-up questions.
>
> ---
>
> > Q1. parameter efficiency, scalability, training cost is likely significant when the number of LLMs scales up (each training query needs to be evaluated on each LLM), and retraining is required when any new LLMs are incorporated.
> >
> > Table 1 lacks a comparison ... training time ... parameter efficiency
>
> **A1.**
> Great suggestions!
> We compare RouterDC with other routing methods on the number of parameters and training time in Table R4 of the `Rebuttal PDF`.
> As shown, all methods are very efficient in both computation and parameters, i.e., only require about **28 minutes** of training time and have only **86M parameters**.
>
> Moreover,
> RouterDC is **data-efficient** in training. Though each training query needs to be evaluated on all candidate LLMs, the training set can be very small.
> Figure R2 in the `Rebuttal PDF`
> shows the performance of RouterDC with different numbers of training samples per task.
> We can see that
> the testing accuracy saturates quickly, indicating that **a small number of samples is sufficient** for learning the router.
> Moreover, with only 30 samples per task,
> RouterDC already outperforms the previous SOTA overall (57.21 vs 55.77).
> Thus, the total number of runs to query LLMs is affordable.
>
> As RouterDC requires very little training time, retraining the router when new LLMs are incorporated would not be a practical issue.
> Moreover, learning the router incrementally without retraining is also practical for future work.
>
> We will include computation and data efficiency analysis in the revision.
>
> ---
>
> > Q2. incorporate LLM costs into the loss function.
> >
> > Evaluation results on RouterBench
>
> **A2.**
> Thanks for your suggestions.
> Our primary focus is training a router to select suitable LLMs for queries.
> Hence, performance is used as the main metric for learning the router.
>
> To resolve reviewer's concerns,
> we further conducted experiments on two tasks of RouterBench (i.e., GSM8K and MBPP) and considered the LLM costs.
> We modify the score $s_i^{(t)}$ to $s_i^{(t)}+c_i^{(t)}$, where $c_i^{(t)}$ is the cost of query $x_i$ using the $t$th LLM.
> Figure R3 in the `Rebuttal PDF`
> shows that RouterDC is more cost-effective than CosineClassifier and ZOOTER in both tasks.
>
> ---
>
> > Q3. including incapable LLMs can lead to unnecessary computational overhead and performance degradation.
> > There is a lack of analysis for this issue.
>
> **A3.**
> Thanks for your insightful comments.
> We agree that incapable LLMs are unnecessary.
> For example, Figure 9 shows that very few queries are routed to Chinese-Mistral-7B and dolphin-2.6-mistral-7b,
> thus, removing them can reduce computations without sacrificing performance.
> In practice,
> one can use a hold-out validation to analyze the usage of candidate LLMs and off-load those LLMs that are rarely used.
>
> ---
>
> > Q4. improvements after incorporating sample-sample contrastive loss seem marginal. Can this be further analyzed or explained?
>
> **A4.**
> Sorry for the confusion caused by Figures 5 and 7 due to the large y-axis range.
> In fact,
> RouterDC (w/ $\mathcal{L}\_\text{sample-LLM}$ + $\mathcal{L}\_\text{sample-sample}$) achieves better average accuracy than RouterDC (w/ $\mathcal{L}\_\text{sample-LLM}$) by a significant margin of $2.02\%$ (Table 4 in Appendix A, $\lambda=1$ vs $\lambda=0$).
> We will clarify this in the revision.
>
> ---
>
> > Q5. On OOD tasks, RouterDC performs worse than the best-performing LLM on certain individual tasks. Is there any analysis of the reasons?
>
> **A5.**
> OOD tasks are much more challenging.
> Though RouterDC fails to achieve the highest accuracy on all OOD tasks, RouterDC can assemble complementary abilities of the candidate LLMs and achieve the best overall performance.
> Specifically, RouterDC performs comparably to the best candidate LLMs on all tasks (i.e., 38.81 vs 39.71 for PreAlgebra, 46.80 vs 47.34 for MBPP, and 51.93 vs 52.01 for C-EVAL).
> Moreover, RouterDC outperforms existing routing methods by a large margin.
> We will add this discussion to the revision.
>
> ---
>
> > Q6. comparison of normalized average score.
>
> **A6.** Thanks for your insightful suggestion. We normalize the score of method $\mathbb{A}$ by
> $$\frac{\text{Acc on task t using method $\mathbb{A}$}}{\text{Acc on task t using dolphin-2.9-llama3-8b}} \times 100\\%.$$
> Tables R5 and R6 in the `Rebuttal PDF` report the normalized scores for the ID and OOD scenarios, respectively,
> showing that RouterDC outperforms existing routing methods by a large margin.
>
> ---
>
> > Q7. no need to include RandomSelect
>
> **A7.** Thanks for your suggestion. We will remove it accordingly in the revision.
>
> ---
>
> > Q8. In the "Routing to Different Numbers of LLMs" evaluation, why were LLMs added in the chosen order? Based on Table 1, adding them in performance-descending order might yield smaller accuracy enhancements.
>
> **A8.**
> Thanks for your comments.
> The adding order of LLMs in Figure 13 is the same as the order of candidate LLMs (top to bottom) in Table 1.
> We agree that the order will affect the accuracy improvement, e.g.,
> adding an incapable LLM will yield small or no improvement.
>
> ---
>
> > Q9.
> Would an LLM-dropping mechanism improve overall performance?
>
> **A9.**
> Good suggestion.
> As incapable LLMs are unnecessary,
> one can greedily drop them according to validation performance.
> For example, by dropping Mistral-7B and Chinese-Mistral-7B,
> the average accuracy increases from 58.54 to 58.67.
>
> ---
>
> > Q10. Did you cluster queries within one benchmark or across several benchmarks?
>
> **A10.**
> Sorry for the confusion.
> We cluster all training queries from **several** benchmarks.
> We will clarify this in the revision.
>
> ---
>
> > Q11. broader implications: scalability and cost efficiency
>
> **A11.**
> Please see our reply to Q1 and Q2.

---

> ### Comment · Reviewer_4ok1 · 2024-08-12
>
> Thank you for the detailed responses. I am happy to vote for acceptance. Please ensure that these discussions and results are included in your revision.
>
> However, after re-reading the paper, I noticed that it currently lacks a comparison or at least a discussion of the existing cascade-based approaches [1-4] for assembling LLMs. Unlike ensemble-based methods, cascade approaches do not necessarily invoke all models and can also achieve good cost-performance trade-offs as routing approaches (indeed, they may invoke multiple LLM calls for one query, but this does not necessarily mean that their cost-efficiency is worse). Including such a discussion or comparison would provide a more comprehensive understanding of your work within the context of existing research.
>
> [1] Model Cascading: Towards Jointly Improving Efficiency and Accuracy of NLP Systems, EMNLP 2022
>
> [2] Language Model Cascades: Token-level Uncertainty and Beyond, ICLR 2024
>
> [3] Large Language Model Cascades with Mixture of Thoughts Representations for Cost-efficient Reasoning, ICLR 2024
>
> [4] Online Cascade Learning for Efficient Inference over Streams, ICML 2024

---

> > ### Author Response · Authors · 2024-08-14
> > **Reply to Reviewer 4ok1**
> >
> > Thanks again for your positive rating.
> > We will certainly add the above discussions and experiments to the revision.
> >
> > ---
> >
> > > Q12. However, after re-reading the paper, I noticed that it currently lacks a comparison or at least a discussion of the existing cascade-based approaches [1-4] for assembling LLMs.
> >
> > **A12.**
> > We have discussed a cascade-based method (i.e., FrugalGPT) in the related work of the paper (Lines 75-77, Lines 182-184).
> > Thanks for bringing the four other cascade-based methods to our attention.
> > We agree that cascade-based methods are one direction to achieve cost-effectiveness when choosing LLMs, but they are different from RouterDC in terms of **settings, inference cost, and tasks**.
> > We discuss the differences between our RouterDC and the mentioned cascade-based methods below.
> >
> >
> > (i)
> > RouterDC considers **a different setting** compared with the cascade-based methods [1-4]. The cascade-based methods usually **assume that the capacity of LLMs depends on the model size**.
> > Their intuitive idea is to query LLMs from weak (small) to strong  (large) until a satisfactory answer is obtained instead of calling the strong LLMs for all queries.
> > Our RouterDC does not require this assumption and can select a suitable LLM from multiple small or large candidate LLMs.
> > Hence, routing-based methods are more general.
> > Furthermore, even if LLMs are of the same size, they may have different specialized capabilities.
> >
> > (ii)
> > Cascade-based methods [1-4] may call LLMs **multiple times** for a query (in the worst case, all candidate LLMs need to be called), but our RouterDC only needs to call the selected LLM **once** in inference/testing.
> >
> > (iii)
> > RouterDC is general and can be used for **generation tasks**, but Model Cascading [1] and Online Cascade Learning [4] are limited to **classification task** (e.g., SST2, MRPC, IMDB).
> > Generation tasks are usually more useful and challenging than classification tasks in NLP.
> >
> > We will include the above discussion and related works in the revision.
> >
> > ---
> >
> > #### References
> >
> > [1] Model Cascading: Towards Jointly Improving Efficiency and Accuracy of NLP Systems, EMNLP 2022
> >
> > [2] Language Model Cascades: Token-level Uncertainty and Beyond, ICLR 2024
> >
> > [3] Large Language Model Cascades with Mixture of Thoughts Representations for Cost-efficient Reasoning, ICLR 2024
> >
> > [4] Online Cascade Learning for Efficient Inference over Streams, ICML 2024

---

### Official Review · Reviewer_GGL7 · 2024-07-10

**Soundness:** 3
**Presentation:** 3
**Contribution:** 2
**Rating:** 6
**Confidence:** 4

**Summary:**

The authors propose a routing between different LLMs, based on classification of embeddings from a fine-tuned transformer model (mDeBERTaV3-base), on a per-query basis.  To sharpen the classification, they chose positive and negative samples among the training tasks, and to stabilise training they add a loss term that encourages cohesive clusters.  The results suggest that the system is capable of routing appropriately.

**Strengths:**

* Experiments use a realistic pool of local LLMs
* Practical to implement - though makes most sense if the upstream LLMs are being paid for per-token, rather than by availability

Original Rating: 4
Original Confidence: 3

**Weaknesses:**

* Since the LLMs in the pool appear to be pretty decisively good at some of the tasks (Figure 9), the meat of the router task is to see whether it's possible to classify these different tasks via a small LM and embeddings.  On its face, it doesn't seem too remarkable that this works.
* Sample-Sample Contrastive Loss : Smells like a post-experiment fix, rather than a principled choice
  + Particularly since a pretrained (frozen?) mDeBERTaV3-base is used to determine which samples 'belong' to which clusters

Minor point(s)

* L30: "Figure 3 shows the scores of seven LLMs for an example query, where the top three LLMs have significantly higher scores."
  + Seems unclear whether top-3 is so different from top-4 : Maybe a better example could be used

**Questions:**

* For the "OOD" results, how disjoint (really) are (i) CMMLU and C-EVAL; (ii) HumanEval and MBPP?

* Table 3: Robustness of RouterDC to LLM losses during inference - isn't this proving that (for instance) Meta-Llama3-8B is being consistently chosen for CMMLU?

* L46: "To improve the training stability, we cluster the training queries" - intra-group vs inter-group (within batch?)
  + L225: The number of clusters N is set to 5 : Is it a coincidence that this matches the number of training tasks?
    - Appendix B : Is it troubling that task-identity is roughly the same as pretrained cluster labels?  Perhaps the task prompts are a give-away...
    - L257: "Moreover, increasing N leads to higher average accuracy when N is small (≤ 4), but the accuracy saturates quickly." (ditto)

* How could this approach work in a chat context?  Is it just for single queries?

**Limitations:**

* The router being learned here is a task-wise classifier.  There is likely commercial value in having this kind of routing, but implementing/testing this doesn't seem particularly novel (apologies for the vagueness).

---

> ### Author Rebuttal · Authors · 2024-08-07
>
> Thanks for your efforts and useful comments.
> We take all comments seriously and hope that our reply can resolve your concerns.
> Please let us know if you have any follow-up questions.
>
> ---
>
> > Q1: the meat of the router task is to see whether it's possible to classify these different tasks via a small LM and embeddings
> >
> > The router being learned here is a task-wise classifier.
>
> **A1.**
> Sorry for the confusion.
> Though Figure 9 shows that RouterDC can route most of the queries from the same task to the same LLM, RouterDC is a query-based router instead of a task-wise classifier.
>
> - RouterDC is **not a task-wise classifier** that selects an LLM for each task. The performance of a task-wise classifier is bounded by that of the top-performing LLM.
> However, Table 1 shows that RouterDC can beat the top-performing LLMs on GSM8K, ARC-C, and HumanEval, suggesting that RouterDC is not simply a task-wise classifier. Furthermore, Figure 9 shows that RouterDC does not always route all queries from the same task to the same LLM. For example, RouterDC assigns 96% and 4% of GSM8K queries to MetaMath-Mistral-7B and dolphin-2.9-llama3-8b, respectively.
> - RouterDC is **a query-based router.**  All training queries are merged together to learn the router. At the testing stage, the learned router assigns the testing query to the suitable LLM based on the similarity between the query and LLM embeddings. Both sample-LLM and sample-sample losses do not require the task identity.
> - The previous work **LoraRetriever (ACL 2024) is exactly a task-wise classifier**. As the task identity is unavailable in practice, the cluster label is used instead. Tables 1 and 2 show that LoraRetriever (clustering) is worse than RouterDC, indicating that RouterDC routes queries more effectively.
>
> ---
>
> > Q2: Sample-Sample Contrastive Loss : Smells like a post-experiment fix, rather than a principled choice.
> > ...
> > mDeBERTaV3-base is used to determine which samples 'belong' to which clusters
>
> **A2.**
> We agree that the sample-sample loss is designed to deal with the training instability observed in early experiments.
> Contrastive learning is an effective technique to retain the semantic similarity of sentences in the embedding space (SimCSE, EMNLP 2021; Sentence-T5, ACL 2022).
> Hence,
> we introduce the sample-sample loss to encourage the encoder to generate similar embeddings for semantically similar queries.
>
> Note that at inference (testing), RouterDC does not need to cluster queries.
>
> ---
>
> > Q3.
> Seems unclear whether top-3 is so different from top-4.
>
> **A3.**
> Thank you for pointing it out.
> We will fix it in the revision: "the top-three LLMs have significantly higher scores than the bottom-three LLMs".
>
> ---
>
> > Q4. how disjoint (really) are (i) CMMLU and C-EVAL; (ii) HumanEval and MBPP?
>
>
> **A4.**
> Thanks for the question.
> (i) A detailed comparison between **CMMLU and C-EVAL** is given in Appendix A of the CMMLU paper (arXiv:2306.09212), showing that
> they have different distributions and contain only 74 shared samples (**about 1%** of the CMMLU dataset).
> (ii) We check the overlap between **HumanEval and MBPP** by string matching and find that they are **completely disjoint**.
>
> ---
>
> > Q5. isn't this proving that (for instance) Meta-Llama3-8B is being consistently chosen for CMMLU?
>
> **A5.**
> Yes, Meta-Llama3-8B is consistently chosen for CMMLU queries unless missing since it performs significantly better than other candidate LLMs on CMMLU (Table 1),
> confirming that the learned router can route queries to suitable LLMs.
>
> ---
>
> > Q6. L46: "we cluster the training queries" - intra-group vs inter-group (within batch?)
>
> **A6.**
> Sorry for the confusion.
> We cluster **all** the training queries into several groups.
> At each iteration, for a query, we sample its in-group query and out-group queries from the same mini-batch.
> We will clarify this in the revision.
>
> ---
>
> > Q7. Is it a coincidence that N=5 matches the number of training tasks?
>
> **A7.**
> Sorry for the confusion.
> The number of clusters $N$ does not have to be the number of training tasks.
> We have conducted an experiment in the paper to study the sensitivity of $N$ (Lines 255-258).
> As shown in Figure 6, **RouterDC is insensitive to a wide range of $N\in [4, 9]$**, where the number of tasks is 5.
> In practice,
> we can choose
> $N$ by grid search using K-fold cross-validation.
> We will clarify this in the revision.
>
> ---
>
> > Q8.  Appendix B : Is it troubling that task-identity is roughly the same as pretrained cluster labels? Perhaps the task prompts are a give-away.
> >
> > "increasing N ... saturates quickly." (ditto)
>
> **A8.**
> Sorry for the confusion.
> To study the relationship between task-identity and cluster labels, we construct their confusion matrix in Table R1 of the `Rebuttal PDF`.
> As shown, some task identities are different from cluster labels.
> For example, the HumanEval queries are grouped into three clusters.
> We also construct the confusion matrix when $N=4$
> and $N=9$ in Tables R2 and R3 of the `Rebuttal PDF`, respectively.
> Again, queries from the same task can be grouped into different clusters and a cluster can be shared across different tasks.
> We will add this discussion to the revision.
>
>
> ---
>
> > Q9. How could this approach work in a chat context? Is it just for single queries?
>
> **A9.**
> Great suggestion!
> Though RouterDC is designed as a query-based router,
> the framework can be extended to the chat context, e.g., selecting LLMs based on the recent conversation. We will study this in our future work.
>
> ---
>
> > Q10. doesn't seem particularly novel
>
> **A10.**
> Novelty of RouterDC includes **two contrastive losses**: (i) the sample-LLM loss pulls the query embeddings closer to the embeddings of the top-performing LLMs while pushing them away from the embeddings of the bottom-performing LLMs; and
> (ii) the sample-sample loss for training stability.
>
> The novelty is also recognized by
> Reviewers cnpT (**novel**) and
> 4ok1 (**innovative**).

---

> > ### Comment · Reviewer_GGL7 · 2024-08-11
> >
> > * A1+A8 - It was already clear that you are doing query-based rather than task-based classification.  One of my concerns was that for the data you trained/tested over, the two things were so closely matched that the different datasets would leak the task just due to the wording, etc.  Your Table R2 helps.
> >
> > * A4 : I'm surprised that there are *any* strict overlaps between the datasets : Interesting!  But that doesn't change the point that the the claim that the alternate datasets are 'OOD' seems like an overreach.
> >
> > I'm happy to update a little:
> >
> > New Rating: 5
> > New Confidence : 4

---

> > > ### Author Response · Authors · 2024-08-12
> > > **Reply to Reviewer GGL7**
> > >
> > > Thanks for your further comments and **raising the score**.
> > > For the remaining concerns, we address them as follows.
> > >
> > > ---
> > >
> > > > Q11. One of my concerns was that for the data you trained/tested over, the two things were so closely matched that the different datasets would leak the task just due to the wording, etc.
> > >
> > > **A11.**
> > > We understand the reviewer’s concern that certain words in the query may leak the task identity, making it easy for RouterDC to perform like a task classifier.
> > >
> > > To resolve this concern,
> > > we conducted an additional experiment in **a single task setting**, i.e., we train the router on the training set of HumanEval and evaluate it on the testing set.
> > > The
> > > single task setting is an edge case where **all queries may contain the same task information**.
> > > Hence,
> > > the router needs to learn how to route queries appropriately based on the query itself instead of some possible task information contained in the query.
> > > Table below reports the testing accuracy.
> > > As can be seen,
> > > **RouterDC largely outperforms the best candidate LLM (i.e., dolphin-2.9-llama3-8b) and existing routing methods**,
> > > demonstrating that the router can select appropriate LLMs for queries based on query characteristics.
> > > We will add the experiment and discussion to the revision, which definitely improve our work.
> > >
> > > \begin{array}{lc}
> > > \hline
> > > \text{Method} &   \text{HumanEval}   \newline
> > > \hline
> > > \text{Mistral-7B} & 28.98 \newline
> > > \text{MetaMath-Mistral-7B} & 29.80 \newline
> > > \text{zephyr-7b-beta} & 22.04 \newline
> > > \text{Chinese-Mistral-7B} & 21.43 \newline
> > > \text{dolphin-2.6-mistral-7b} & 45.10 \newline
> > > \text{Meta-Llama-3-8B} & 26.73 \newline
> > > \text{dolphin-2.9-llama3-8b} & 49.39 \newline
> > > \hline
> > > \text{ZOOTER} & 39.38 \newline
> > > \text{CosineClassifier} & 52.45 \newline
> > > \text{RouterDC} & \mathbf{56.32} \newline
> > > \hline
> > > \end{array}
> > >
> > > ---
> > >
> > > > Q12. I'm surprised that there are any strict overlaps between the datasets : Interesting!
> > >
> > > **A12.**
> > > Thanks for your comments!
> > > We guess there was a typo in the comment, it should be "there are **NOT** any strict overlaps between the datasets : Interesting!"
> > >
> > >
> > > > Q13. But that doesn't change the point that the the claim that the alternate datasets are 'OOD' seems like an overreach.
> > >
> > > **A13.**
> > > Thanks for your insightful comments!
> > > We appreciate the reviewer raising the concern about the definition of OOD.
> > > In the paper, C-EVAL and MBPP are treated as OOD tasks as they are different task distributions or question-answer instruction.
> > > We briefly summarize their differences below.
> > >
> > > (i) CMMLU and C-EVAL have **different task distributions**.
> > > CMMLU contains more culture- and region-related tasks, while C-EVAL have more STEM tasks.
> > > Moreover, CMMLU and C-EVAL use **different prompts** to ask multiple-choice question.
> > > C-EVAL uses continuous underscores to indicate the answer's location, whereas CMMLU employs no special notation for referencing the answer, except for brackets when the answer is within a sentence.
> > >
> > > (ii)
> > > HumanEval and MBPP assess the code generation proficiency of LLM from **two distinct perspectives**.
> > > A HumanEval query **gives the header** of a python function and some comments, requiring the LLM to **implement the rest** of the function. On the other hand, a MBPP query **gives an intent** and  asks the LLM to **generate the function from scratch**.
> > >
> > > To further resolve this concerns,
> > > we evaluate the learned router on **one more OOD task: JavaScript** [R1],
> > > which aims to generate JavaScript code to solve problems.
> > > Different from HumanEval, which generates Python code to solve problems,
> > > JavaScript can be viewed as **a distant OOD task**.
> > > Table below reports the testing accuracy.
> > > As can be seen, RouterDC outperforms existing routing methods by a large margin, demonstrating
> > > that our RouterDC is more effective in routing queries of the distant OOD task.
> > >
> > > We will include the additional experiments and discussions in the revision.
> > >
> > >
> > > \begin{array}{lc}
> > > \hline
> > >  &   \text{JavaScript}   \newline
> > > \hline
> > > \text{Mistral-7B} & 29.88 \newline
> > > \text{MetaMath-Mistral-7B} & 31.83 \newline
> > > \text{zephyr-7b-beta} & 11.71 \newline
> > > \text{Chinese-Mistral-7B} & 17.68 \newline
> > > \text{dolphin-2.6-mistral-7b} & 45.00 \newline
> > > \text{Meta-Llama-3-8B} & 37.07 \newline
> > > \text{dolphin-2.9-llama3-8b} & 53.84 \newline
> > > \hline
> > > \text{ZOOTER} & 41.64 \newline
> > > \text{CosineClassifier} & 37.32 \newline
> > > \text{RouterDC} & \mathbf{48.66} \newline
> > > \hline
> > > \end{array}
> > >
> > > ---
> > >
> > > ### references
> > >
> > > [R1] CodeGeeX: A Pre-Trained Model for Code Generation with Multilingual Benchmarking on HumanEval-X. KDD 2023.

---

> > > > ### Comment · Reviewer_GGL7 · 2024-08-13
> > > >
> > > > A12. """Thanks for your comments! We guess there was a typo in the comment, it should be "there are NOT any strict overlaps between the datasets : Interesting!" """
> > > >
> > > > Actually, there was no typo.  Your claim is that one dataset is OOD compared to the other : Surely then, it should be surprising that they have textually identical pairs in them?
> > > >
> > > > A11(+A13) : This is good additional work, that goes far further in proving your case.  Thank you.
> > > >
> > > > And for that part, I'll inch up one (last) time:
> > > >
> > > > New Rating: 6  New Confidence : 4

---

> > > > > ### Author Response · Authors · 2024-08-14
> > > > > **Reply to Reviewer GGL7 (2)**
> > > > >
> > > > > We are glad that our additional experiments have resolved your concerns, and we will add the experiments to the revision.
> > > > >
> > > > > ---
> > > > >
> > > > > > Q14. Actually, there was no typo. Your claim is that one dataset is OOD compared to the other : Surely then, it should be surprising that they have textually identical pairs in them?
> > > > >
> > > > > **A14.**
> > > > > We apologize for our misunderstanding of your comments.
> > > > > As discussed previously, most of the CMMLU and C-EVAL samples (about 99%) are different. Thus, the tiny overlap may not significantly affect the evaluation.

---

### Official Review · Reviewer_cnpT · 2024-07-12

**Soundness:** 4
**Presentation:** 3
**Contribution:** 3
**Rating:** 7
**Confidence:** 4

**Summary:**

This paper studies the problem of assembling off-the-shelf LLMs to harness their complementary strengths. The authors propose a novel query-based router by Dual Contrastive learning (RouterDC), i.e., a sample-LLM contrastive loss and a sample-sample contrastive loss. The former contrastive loss aims at training the router such that it can assign suitable LLMs for queries, while the latter is for training stability. Experiments on various challenging tasks demonstrate that RouterDC performs better than existing routing methods and individual top-performing LLMs in both in-distribution and out-distribution settings.

**Strengths:**

1. The proposed dual contrastive losses for training a query-based router are novel. The sample-LLM contrastive loss seems more sound for learning the router than the KL loss used in previous work ZOOTER.
2. Extensive experiments on both in-distribution and out-distribution tasks show that the proposed RouterDC outperforms existing routing methods on average. The performance shown in Fig.2 confirms that RouterDC can harness the complementary strengths of off-the-shelf LLMs.
3. The paper is well-written and easy to follow. It provides many comprehensive ablation experiments, e.g., Fig. 11 shows the advantage of the sample-sample contrastive loss in improving the training stability of RouterDC.

**Weaknesses:**

1. In Line 157, the authors claim that “The reason is that some similar queries can have dissimilar embeddings and may be routed to different LLMs.” Evidence should be provided to support this claim.
2. For the OOD setting (Table 2), RouterDC fails to beat the best individual LLMs on all tasks (e.g., 38.81 vs 39.72 on PreAlgebra).
3. RouterDC may require a large amount of labeled data to train the router.

**Questions:**

1. How about the performance of RouterDC without the sample-sample contrastive loss?
2. In Fig. 9, it seems that no samples are routed to the Chinese-Mistral-7B model, why?
3. Can you visualize the embeddings of training samples extracted by the encoder $\mathcal{E}(x;w)$ of RouterDC?

**Limitations:**

The authors are encouraged to discuss limitations in the conclusion section.

---

> ### Author Rebuttal · Authors · 2024-08-07
>
> We sincerely thank you for the detailed and positive comments.
> We take all comments seriously and do our best to address every concern raised.
> Please let us know if you have any follow-up questions.
>
> > Q1. In Line 157, the authors claim that “The reason is that some similar queries can have dissimilar embeddings and may be routed to different LLMs.” Evidence should be provided to support this claim.
>
> **A1.**
> Thanks for your suggestion.
> Figure R1(a) in the `Rebuttal PDF` shows t-SNE visualization of training query embeddings extracted by the encoder trained by $\mathcal{L}\_\text{sample-LLM}$.
> As can be seen, query embeddings belonging to different tasks are roughly mixed together.
> We also provide two GSM8K queries as follows,
> which require basic calculation of shopping costs.
> Their embeddings have very low similarity (only $-0.4589$)
> when the router is trained by $\mathcal{L}\_\text{sample-LLM}$ alone.
>
>
>
> ```
> Mary does her grocery shopping on Saturday. She does her shopping only at a specific store where she is allowed a credit of $100, which must be paid in full before her next shopping trip. That week she spent the full credit limit and paid $15 of it on Tuesday and $23 of it on Thursday. How much credit will Mary need to pay before her next shopping trip?
> ```
>
> ```
> Betty is saving money for a new wallet which costs $100. Betty has only half of the money she needs. Her parents decided to give her $15 for that purpose, and her grandparents twice as much as her parents. How much more money does Betty need to buy the wallet?
> ```
>
> After integrating $\mathcal{L}\_\text{sample-sample}$,
> training query embeddings have a clear cluster structure (Figure R1(b)).
> Moreover, the similarity between the above queries increases to $0.9982$.
>
> We will add this discussion to the revision.
>
> ---
>
> > Q2. For the OOD setting (Table 2), RouterDC fails to beat the best individual LLMs on all tasks (e.g., 38.81 vs 39.72 on PreAlgebra).
>
> **A2.**
> OOD tasks are much more challenging than id-distribution (ID) tasks.
> Though our RouterDC fails to achieve the highest accuracy on every task, RouterDC can assemble complementary abilities of the candidate LLMs and achieve the best overall performance (an improvement of 1.90%).
> Besides, RouterDC performs comparably to the best candidate LLMs on all tasks (i.e., 38.81 vs 39.71 for PreAlgebra, 46.80 vs 47.34 for MBPP, and 51.93 vs 52.01 for C-EVAL).
> Moreover, RouterDC outperforms existing routing methods by a large margin overall.
> We will add this discussion to the revision.
>
> ---
>
> > Q3. RouterDC may require a large amount of labeled data to train the router.
>
> **A3.**
> To resolve this concern,
> we conducted an experiment to study the performance of RouterDC with different numbers of training samples per task.
> As can be seen from Figure R2 in the `Rebuttal PDF`,
> the testing accuracy saturates quickly, indicating that **a small number of samples is sufficient** for learning the router (e.g., 100 samples per task).
> Moreover, with only 30 samples per task,
> RouterDC already outperforms the previous SOTA overall (57.21 vs 55.77), demonstrating that our RouterDC does not require a large amount of labeled data for training.
> We will include the experiments and data efficiency analysis in the revision.
>
> ---
>
> > Q4. How about the performance of RouterDC without the sample-sample contrastive loss?
>
> **A4.**
> Thanks for the suggestion.
> We compare RouterDC w/ or w/o the sample-sample loss with the previous SOTA in the following tables.
> As can be seen, in both ID and OOD scenarios,
> using the sample-LLM loss alone (i.e., RouterDC (w/ $\mathcal{L}\_\text{sample-LLM}$)) performs better than the previous SOTA (with an average accuracy improvement of $0.75\\%$ in ID scenario and $0.50\\%$ in OOD scenario).
> We will add this discussion to the revision.
>
> \begin{array}{lcccccl}
> \hline
>  \textbf{(in-distribution)} &   \text{MMLU} &   \text{GSM8K} &   \text{CMMLU} &   \text{ARC-C} &   \text{HumanEval} &   \text{Avg} \newline
> \hline
> \text{Preivous SOTA} & 63.33 & 66.63 & 51.77 & 57.10 & 40.00 & 55.77  \newline
> \text{RouterDC } (\text{w/ } \mathcal{L}\_\text{sample-LLM})  & 63.21 & 68.87 & 49.27 & 49.43 & 51.84 & 56.52 \ \text{  (+0.75)}\newline
> \text{RouterDC } (\text{w/ } \mathcal{L}\_\text{sample-LLM}+\mathcal{L}\_\text{sample-sample}) & 61.07 & 70.32 & 51.77 & 58.52 & 51.02 & \mathbf{58.54}\text{ (+2.77)} \newline
> \hline
> \end{array}
>
> \begin{array}{lcccl}
> \hline
>  \textbf{(out-of-distribution)} &   \text{Pre-Algebra} &   \text{MBPP} &   \text{C-EVAL} & \text{Avg} \newline
> \hline
> \text{Preivous SOTA} & 35.36 & 43.12 & 52.01 & 43.50  \newline
> \text{RouterDC } (\text{w/ } \mathcal{L}\_\text{sample-LLM})  & 36.51 & 47.34 & 48.14 & 44.00 \ \text{  (+0.50)}\newline
> \text{RouterDC } (\text{w/ } \mathcal{L}\_\text{sample-LLM}+\mathcal{L}\_\text{sample-sample}) & 38.81 & 46.80 & 51.93 & \mathbf{45.85} \text{ (+2.35)}\newline
> \hline
> \end{array}
>
> ---
>
> > Q5. In Fig. 9, it seems that no samples are routed to the Chinese-Mistral-7B model, why?
>
> **A5.**
> Thanks for your insightful question.
> We can see from Table 1 that Chinese-Mistral-7B is incapable of all tasks and has the worst overall performance,
> suggesting that its specialized ability may be covered by other candidate LLMs.
> Hence, no samples are routed to Chinese-Mistral-7B, which also verifies that RouterDC can select suitable LLMs for queries.
> We will add this discussion to the revision.
>
> ---
>
> > Q6. Can you visualize the embeddings of training samples extracted by the encoder $\mathcal{E}(x;w)$ of RouterDC?
>
> **A6.**
> Thanks for the suggestion.
> Figure R1(b) in the `Rebuttal PDF` shows the t-SNE visualization of training queries extracted by the learned encoder of RouterDC. As can be seen, the embeddings exhibit a clear structure.

---

### Official Review · Reviewer_ytgM · 2024-07-17

**Soundness:** 2
**Presentation:** 3
**Contribution:** 2
**Rating:** 5
**Confidence:** 5

**Summary:**

This paper aims to improve the ability of LLMs by assembling them. The proposed method use a contrastive learning strategy. It uses a sample-LLM contrastive loss which pull the query embedding closer to the top-performed LLM embedding. It also employs a sample-sample contrastive loss, which learns the distribution of input with the help of clustering.

**Strengths:**

The contrastive learning idea is intuitive.

The routing model is only 86m, which is quite small compared to the LLMs. And the method achieve better results compared with simple voting or learning the routing with a reward model.

**Weaknesses:**

The contribution of this work lies in two part, first, it uses sample-llm contrastive loss, instead of directly learning the routing as a classification; second, it use sample-sample contrastive loss to make the training more stable. However, the relation between the two is missing. I am wondering how the two parts contribute to the final performance. Is it possible to use ZOOTER with the sample-sample loss? What do you expect as the result?

I think the response partially addressed my concern. So I raised my score to 5. I do understand there are hyper-parameters that one could control to affect the results, but are the proposed two contrastive losses the most important ones to be consider in this scenario? I would love to see more discussions in the paper.

**Questions:**

See the weakness part.

---

> ### Author Rebuttal · Authors · 2024-08-07
>
> Thanks for the valuable comments.
> We really appreciate your efforts to help us improve our paper.
> We carefully addressed your concerns below and sincerely hope that our reply resolves your concerns.
> Please let us know if you have any follow-up questions.
>
> > Q1. the relation between the two contrastive losses is missing. I am wondering how the two parts contribute to the final performance.
>
> **A1.**
> Thanks for your insightful suggestions.
> The relation between two contrastive losses can be seen in Figure 5 of the paper, which studies the effect of hyperparameter $\lambda$ in Eq. (5) (i.e., $\mathcal{L}\_{\text{sample-LLM}} + \lambda \ \mathcal{L}\_{\text{sample-sample}}$).
> We can see that using two contrastive losses together (i.e., $\lambda=1$) achieves better overall performance than using the sample-LLM contrastive loss alone (i.e., $\lambda=0$).
> Moreover, the overall performance of RouterDC is not so sensitive to a wide range of $\lambda \in [0.5, 5]$, making it easier to choose the value of $\lambda$.
>
> To further study the contributions of two contrastive losses to the final performance, we report the detailed results for both in-distribution (ID) and out-of-distribution (OOD) scenarios in the following tables.
> Since the sample-LLM loss provides the supervision signal and is essential for training the router, we focus on comparing RouterDC with and without the sample-sample contrastive loss.
> As can be seen from the below tables,
> RouterDC (w/ $\mathcal{L}\_\text{sample-LLM}$ + $\mathcal{L}\_\text{sample-sample}$) averagely outperforms RouterDC (w/ $\mathcal{L}\_\text{sample-LLM}$) in both scenarios, demonstrating the usefulness of the proposed sample-sample contrastive loss.
>
> Moreover,
> compared with the previous SOTA,
> using the sample-LLM contrastive loss alone (i.e., RouterDC (w/ $\mathcal{L}\_\text{sample-LLM}$)) performs better (with an average accuracy improvement of $0.75\\%$ in the ID scenario and $0.50\\%$ in the OOD scenario),
> while
> RouterDC (w/ $\mathcal{L}\_\text{sample-LLM}$ + $\mathcal{L}\_\text{sample-sample}$) achieves better performance by a large margin of $2.77\\%$ in the ID scenario and $2.35\\%$ in the OOD scenario.
>
> We will add this discussion to the revision.
>
> \begin{array}{lcccccl}
> \hline
>  \textbf{(in-distribution)} &   \text{MMLU} &   \text{GSM8K} &   \text{CMMLU} &   \text{ARC-C} &   \text{HumanEval} &   \text{Avg} \newline
> \hline
> \text{Preivous SOTA} & 63.33 & 66.63 & 51.77 & 57.10 & 40.00 & 55.77  \newline
> \text{RouterDC } (\text{w/ } \mathcal{L}\_\text{sample-LLM})  & 63.21 & 68.87 & 49.27 & 49.43 & 51.84 & 56.52 \ \text{  (+0.75)}\newline
> \text{RouterDC } (\text{w/ } \mathcal{L}\_\text{sample-LLM}+\mathcal{L}\_\text{sample-sample}) & 61.07 & 70.32 & 51.77 & 58.52 & 51.02 & \mathbf{58.54}\text{ (+2.77)} \newline
> \hline
> \end{array}
>
> \begin{array}{lcccl}
> \hline
>  \textbf{(out-of-distribution)} &   \text{Pre-Algebra} &   \text{MBPP} &   \text{C-EVAL} & \text{Avg} \newline
> \hline
> \text{Preivous SOTA} & 35.36 & 43.12 & 52.01 & 43.50  \newline
> \text{RouterDC } (\text{w/ } \mathcal{L}\_\text{sample-LLM})  & 36.51 & 47.34 & 48.14 & 44.00 \ \text{  (+0.50)}\newline
> \text{RouterDC } (\text{w/ } \mathcal{L}\_\text{sample-LLM}+\mathcal{L}\_\text{sample-sample}) & 38.81 & 46.80 & 51.93 & \mathbf{45.85} \text{ (+2.35)}\newline
> \hline
> \end{array}
>
> ---
>
> > Q2. Is it possible to use ZOOTER with the sample-sample loss? What do you expect as the result?
>
> **A2.**
> Thanks for your valuable suggestion!
> We conducted additional experiments to study whether the proposed sample-sample contrastive loss is useful for
> ZOOTER.
> The following tables show the testing accuracy for the ID and OOD scenarios.
> As can be seen, integrating $\mathcal{L}\_\text{sample-sample}$ into ZOOTER **leads to improvements** of $+1.52\\%$ and $+0.81\\%$ for ID and OOD, respectively, demonstrating that the proposed sample-sample contrastive loss is **beneficial** for ZOOTER.
> We will include
> the experiments and add this discussion to the revision, which will definitely improve our paper.
>
> \begin{array}{lcccccl}
> \hline
>  \textbf{(in-distribution)} &   \text{MMLU} &   \text{GSM8K} &   \text{CMMLU} &   \text{ARC-C} &   \text{HumanEval} &   \text{Avg} \newline
> \hline
> \text{ZOOTER} & 60.48 & 66.69 & 45.27 & 53.13 & 44.29 & 53.97  \newline
> \text{ZOOTER } (\text{w/ } \mathcal{L}\_\text{sample-sample}) & 60.15 & 69.71 & 46.59 & 54.26 & 46.73 & \mathbf{55.49}\text{ (+1.52)}\newline
> \hline
> \end{array}
>
>
> \begin{array}{lcccl}
> \hline
>  \textbf{(out-of-distribution)} &   \text{Pre-Algebra} &   \text{MBPP} &   \text{C-EVAL} & \text{Avg} \newline
> \hline
> \text{ZOOTER} & 34.44 & 41.10 & 44.95 & 40.16  \newline
> \text{ZOOTER } (\text{w/ } \mathcal{L}\_\text{sample-sample})  & 36.05 & 39.84 & 47.03 & \mathbf{40.97} \ \text{  (+0.81)}\newline
> \hline
> \end{array}

---

> > ### Author Response · Authors · 2024-08-12
> > **A Gentle Reminder for Reviewer ytgM**
> >
> > Dear Reviewer ytgM,
> >
> > We sincerely thank you again for your effort to improve our work.
> > We have provided a detailed response to resolve your concerns.
> > We would like to kindly remind the reviewer that the close date of the reviewer-author discussion is approaching.
> > Please let us know if you have any additional questions or comments.
> >
> > Best,
> >
> > The Authors

---

### Author Rebuttal · Authors · 2024-08-07

Dear Reviewers and ACs,

We sincerely thank all the reviewers and ACs for your insightful and valuable comments.
We are delighted that reviewers find that:
- our work **addresses a significant challenge** in LLM utilization  (`Reviewer 4ok1`).
- our method is **intuitive** (`Reviewer ytgM`), **novel/innovative** (`Reviewers cnpT and 4ok1`), and **practical** (`Reviewer GGL7`).
- RouterDC outperforms existing routing methods (`Reviewers ytgM, cnpT,
 and 4ok1`) and is capable of routing appropriately (`Reviewer GGL7`).

The `Rebuttal PDF` contains the figures and tables that are referred to in the response to reviewers.

Best,

The Authors

---

### Decision · Program_Chairs · 2024-09-25

**Decision:**

Accept (poster)

**Comment:**

Paper proposes RouterDC, a novel query-based routing model for assembling multiple LLMs using dual contrastive learning. While the paper has merits (see below), certain limitations (e.g., the added value of the proposed loss, the "generalizatibility" of the OOD generalization claims, discussion on cascade based methods) make it more suitable as a poster presentation.

The problem is quite relevant, addressing the challenege of effectively leveraging the diverse strengths of multiple LLMs which is becoming a prevalent approach both by the practitioners (with applications that are facing users) and the researchers; THe approach is novel, sample-LLM contrastive loss; results are strong with detailed discussions during rebuttals by authors providing comprehensive and convincing responses.

Also, further emphasizing the data efficiency -- given the concerns about the scalability and cost of training multiple LLMs -- could be considered as an improvement.